# COMBINATORIAL RISING BANDITS

**Seockbean Song[1]\*, Youngsik Yoon[2]\*, Siwei Wang[3], Wei Chen[3], Jungseul Ok[1, 2]†**

[1]Graduate School of Artificial Intelligence, POSTECH, Pohang, Korea
[2]Department of Computer Science and Engineering, POSTECH, Pohang, Korea
[3]Microsoft Research Asia, Beijing, China

{shinebobo, jungseul}@postech.ac.kr

## ABSTRACT

Combinatorial online learning is a fundamental task for selecting the optimal action (or super arm) as a combination of base arms in sequential interactions with systems providing stochastic rewards. It is applicable to diverse domains such as robotics, social advertising, network routing, and recommendation systems. In many real-world scenarios, we often encounter rising rewards, where playing a base arm not only provides an instantaneous reward but also contributes to the enhancement of future rewards, *e.g.*, robots improving through practice and social influence strengthening in the history of successful recommendations. Crucially, these enhancements may propagate to multiple super arms that share the same base arms, introducing dependencies beyond the scope of existing bandit models. To address this gap, we introduce the Combinatorial Rising Bandit (CRB) framework and propose a provably efficient and empirically effective algorithm, Combinatorial Rising Upper Confidence Bound (CRUCB). We empirically demonstrate the effectiveness of CRUCB in realistic deep reinforcement learning environments and synthetic settings, while our theoretical analysis establishes tight regret bounds. Together, they underscore the practical impact and theoretical rigor of our approach. Our code is available at https://github.com/ml-postech/Combinatorial-Rising-Bandits.

## 1 INTRODUCTION

Combinatorial online learning studies how to select an optimal action (super arm) composed of multiple sub-actions (base arms). This formulation captures the structure of many practical decision-making problems, including robotics (Xu et al., 2025; Wakayama & Ahmed, 2024), social advertising (Ge et al., 2025), automatic machine learning (Huang et al., 2021), network routing (Lagos et al., 2025), and recommendation systems (Atalar & Joe-Wong, 2025; Zhu & Van Roy, 2023).

However, previous studies of combinatorial online learning have largely neglected the presence of a *rising* reward nature in practice, where pulling a base arm not only yields an immediate reward but also enhances future rewards. For example, in robotic planning, hierarchical approaches tackle complex tasks by decomposing them into low-level skills, such as grasping and lifting, which act as sub-actions, while the full sequence of these skills constitutes an action. As these low-level skills are reused across different plans, their performance improves (Jansonnie et al., 2024; Mao et al., 2025), reflecting the rising reward nature. Additional real-world scenarios are presented in Appendix A.

A complementary line of work studies rising bandits, where the expected reward of an arm enhances each time the arm is pulled (Fiandri et al., 2024a; Genalti et al., 2024; Heidari et al., 2016; Metelli et al., 2022). However, these studies consider only non-combinatorial settings and, therefore, ignore the structural dependencies that arise when different super arms share base arms. Such overlap couples reward dynamics and makes the problem substantially more complex: while repeatedly pulling a single arm is optimal in the non-combinatorial setting (Heidari et al., 2016), characterizing an optimal policy in the combinatorial regime is far more intricate. A detailed comparison with existing rising bandit formulations is provided in Appendix B.

---

\*Equal contribution
†Corresponding author: Jungseul Ok <jungseul@postech.ac.kr>

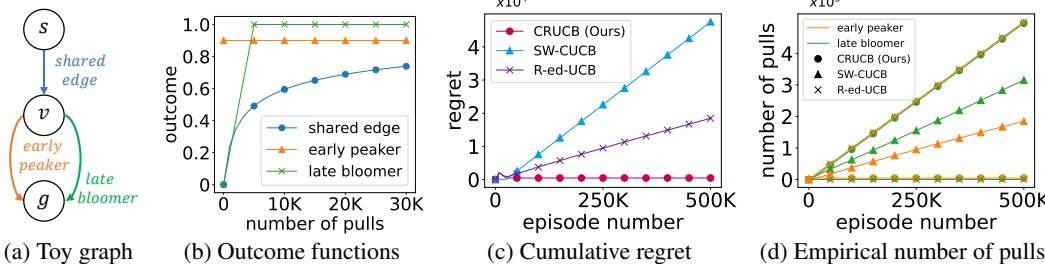

Figure 1: **Toy example for online shortest path planning.** (a) Graph: two paths from $s$ to $g$, an *early peaker path* ({shared edge, early peaker}) and a *late-bloomer path* ({shared edge, late bloomer}). (b) Outcome functions: a *shared edge* rises slowly; *early peaker* starts high but flattens; a *late bloomer* starts low but rises quickly, eventually surpassing the early peaker, so the late bloomer path is optimal for long horizon $T$. The reward is the sum of the outcomes of the base arms. (c) Cumulative regret under three algorithms: CRUCB (ours); SW-CUCB (Chen et al., 2021) (combinatorial bandits); R-ed-UCB (Metelli et al., 2022) (rested rising bandits). CRUCB becomes nearly flat, while SW-CUCB and R-ed-UCB accumulate linear regret. (d) Empirical number of pulls of each edge: CRUCB pulls entirely the late bloomer, SW-CUCB the early peaker, and R-ed-UCB splits pulls roughly evenly.

To model such scenarios, we propose the **C**ombinatorial **R**ising **B**andit (CRB) framework. In this framework, (i) the policy selects a super arm (a set of base arms), and (ii) the outcome of each base arm follows a (rested) rising nature such that the expected outcome of a base arm increases after pulling it as part of the selected super arm. We emphasize that CRB addresses a unique problem that combinatorial bandits (Chen et al., 2013) and rising bandits (Heidari et al., 2016; Metelli et al., 2022) cannot address. While individual base arms behave like rising bandits, super arms do not: shared base arms create dependencies across super arms, leading to *partially shared enhancement*. Figure 1 shows an illustrative example. As depicted in Figure 1d, our proposed algorithm, CRUCB, rapidly converges to selecting the late bloomer path, whereas SW-CUCB (Chen et al., 2021), a combinatorial bandit algorithm, consistently selects the early peaker path due to its inability to account for the rising nature. R-ed-UCB (Metelli et al., 2022), a rising bandit algorithm that ignores the combinatorial structure, splits pulls between both paths because it incorrectly interprets cumulative increments from repeatedly pulling the shared arm as immediate growth, causing it to hallucinate ongoing potential in the early peaker path. This partially shared enhancement distinguishes CRB from prior formulations, introducing fundamentally new challenges. Indeed, this difference also leads to a different characterization of optimality in CRB.

To address the challenges introduced by the partially shared enhancement in CRB, we propose **C**ombinatorial **R**ising **UCB** (CRUCB), a provably efficient algorithm. CRUCB employs a Future-UCB index that optimistically estimates the future outcome of each base arm by combining its recent mean, slope, and uncertainty term, and then solves a combinatorial optimization problem using these estimates of future rewards to select the super arm. On the theoretical side, we derive a regret upper bound for CRUCB and a regret lower bound for CRB, and show that these bounds are close, demonstrating the near-optimal efficiency of our approach. On the empirical side, we conduct extensive experiments comparing CRUCB with a set of baselines in both synthetic environments and deep reinforcement learning tasks, training a neural agent for navigation. These results consistently highlight the superiority of CRUCB and its ability to handle challenges that existing approaches cannot. Therefore, our study positions CRUCB at the intersection of theory and practice: it not only provides provable guarantees but also exposes the limitations of prior methods in realistic environments and demonstrates how CRUCB effectively overcomes them.

Our main contributions are summarized as follows:

- We introduce the Combinatorial Rising Bandit (CRB) framework in Section 2 to formalize rising reward dynamics in combinatorial settings. Furthermore, we analyze the structure of optimal policies, highlighting that CRB differs from prior frameworks and makes the characterization of optimality both intractable and more intricate in Section 3.

- We propose Combinatorial Rising UCB (CRUCB), a provably efficient algorithm for CRB in Section 4, and provide a regret upper bound that nearly matches a corresponding regret lower bound, demonstrating its theoretical tightness in canonical settings in Section 5.

- We extensively validate CRUCB in both synthetic and deep reinforcement learning environments in Section 6. It confirms that CRUCB effectively overcomes the difficulties of the combinatorial rising structure left unsolved by prior methods, while maintaining robustness in practical settings beyond theoretical assumptions.

## 2  PROBLEM FORMULATION

We study the **C**ombinatorial **R**ising **B**andit (CRB) problem, where the mean outcome of each base arm increases with the number of plays. Let $K$ be the number of base arms, $[K] := \{1, \ldots, K\}$, and $\mathcal{S} \subseteq 2^{[K]}$ the set of valid super arms. At each round $t$, a super arm $S_t \in \mathcal{S}$ is chosen, and each $i \in S_t$ yields an outcome $X_i(t)$ drawn independently from a distribution $D_i(N_{i,t-1})$, where $N_{i,t-1}$ is the number of past plays of arm $i$ up to $t - 1$. We assume that $D_i(n)$ is $\sigma^2$-subgaussian with known $\sigma$, and define $\mu_i(n) := \mathbb{E}_{X \sim D_i(n)}[X]$, where $\mu_i(n) \in [0, 1]$ for all $i, n$. The rising condition requires:

$$\gamma_i(n) := \mu_i(n + 1) - \mu_i(n) \geq 0, \quad \forall i \in [K], n \geq 1. \tag{1}$$

Given a chosen super arm $S_t$ and the outcome vector $\boldsymbol{X}_t = \{X_i(t) : i \in S_t\}$, the reward is $R_t := R(S_t, \boldsymbol{X}_t)$, where $R$ is a fixed function. We consider a canonical setting of semi-bandit feedback at time $t$, i.e., $\pi$ selects super arm $S_t$ based on history $\mathcal{F}_{t-1} := \{(S_{t'}, \boldsymbol{X}_{t'}) : t' \in [t - 1]\}$.

For analytical tractability, we assume the concavity of $\mu_i$ as in Heidari et al. (2016):

**Assumption 1.** (Concavity of $\mu_i$) For each $i \in [K]$ and $n \geq 1$, we have $\gamma_i(n) \geq \gamma_i(n + 1)$.

We further assume the monotonicity of the reward function, which is canonical in combinatorial bandit literature (Chen et al., 2016; Wang & Chen, 2018; Wang et al., 2023):

**Assumption 2** (Monotone reward). For each super arm $S \in \mathcal{S}$, the expected reward can be expressed as a function of the mean outcomes of its base arms. Formally, there exists a function $r$ such that

$$\mathbb{E}[R(S, \boldsymbol{X})] = r(S, \boldsymbol{\mu}), \quad \boldsymbol{\mu} = \{\mu_i : i \in S\}, \tag{2}$$

where $\boldsymbol{X}$ denotes the outcome vector. Moreover, $r$ is monotone: for any $S \in \mathcal{S}$ and vectors $\boldsymbol{\mu}, \boldsymbol{\mu}'$ with $\mu_i \leq \mu_i'$ for all $i \in S$, we have $r(S, \boldsymbol{\mu}) \leq r(S, \boldsymbol{\mu}')$. Additionally, we assume $r(S, \boldsymbol{0}) = 0$.

We note that this assumption is verified by various choices of reward functions such as the additive function (Combes et al., 2015; Kveton et al., 2015) and $k$-MAX function (Wang et al., 2023).

**Regret minimization**  For a policy $\pi$, its expected cumulative reward over horizon $T$ is $\mathbb{E}_\pi\left[\sum_{t \in [T]} R_t\right]$. Let $\pi^* := \arg\max_\pi \mathbb{E}_\pi\left[\sum_{t \in [T]} R_t\right]$ be the optimal policy. Then the regret of $\pi$ is defined as:

$$\text{Reg}(\pi, T) := \mathbb{E}_{\pi^*}\left[\sum_{t \in [T]} R_t\right] - \mathbb{E}_\pi\left[\sum_{t \in [T]} R_t\right], \tag{3}$$

where we want to design $\pi$ minimizing this.

## 3  CHARACTERIZATION OF OPTIMALITY

We first study the structure of the optimal policy for CRB. Our key finding is that although the optimal policy is complex in general, a constant policy, which constantly plays the same super arm, can often serve as an effective and even optimal strategy under mild assumptions. We begin with a formal definition of optimal constant policy, which is the best among all possible constant policies.

**Definition 1** (Optimal constant policy). For any super arm $S \in \mathcal{S}$, let $\pi^S$ denote the constant policy that selects $S$ at every round, i.e., $\pi^S(t) = S$, for each $t \in [T]$. The optimal constant policy is $\pi^*_{\text{const}} := \pi^{S^*_{\text{const}}}$, where $S^*_{\text{const}} = \arg\max_{S \in \mathcal{S}} \mathbb{E}_{\pi^S}[\sum_{t \in [T]} R_t]$.

We note that $\pi^*_{\text{const}}$ is optimal in non-combinatorial rising settings (Heidari et al., 2016), which are special instances of CRB such that $\mathcal{S} = \{\{1\}, \{2\}, ..., \{K\}\}$. However, we found that in general, $\pi^*_{\text{const}}$ is not (exactly) optimal:

**Theorem 1.** *Under Assumptions 1 and 2, there exists an instance of CRB in which $\pi_{const}^*$ is not optimal.*

The proof is provided in Appendix C.1. As shown in the proof, the optimal policy may begin with a combination of *early peakers* and *late bloomers* (as introduced in Figure 1), before eventually selecting a pure combination of late bloomers to maximize long-term rewards. This implies the optimal policy can be more complex than constant policies due to the partially shared enhancement.

As such, $\pi_{const}^*$ is not exactly optimal in CRB. However, it can still serve as a good approximation under mild assumptions. In particular, if the reward function satisfies additive-bounded reward assumption, which encompasses important reward functions such as additive and $k$-MAX rewards, $\pi_{const}^*$ achieves a cumulative reward close to that of the overall optimal policy.

**Theorem 2.** *Assume that the reward function $r$ is bounded above and below by an additive function for each super arm $S \in \mathcal{S}$ and any $\boldsymbol{\mu} \in [0, 1]^{|S|}$:*

$$B_L \sum_{i \in S} \mu_i \ \leq \ r(S, \boldsymbol{\mu}) \ \leq \ B_U \sum_{i \in S} \mu_i, \tag{4}$$

*where $B_L$ and $B_U$ are non-negative constants.*

*Then, under Assumptions 1 and 2, the cumulative reward ratio of the optimal constant policy $\pi_{const}^*$ to the optimal policy $\pi^*$ is bounded as*

$$\frac{\mathbb{E}_{\pi^*}\left[\sum_{t \in [T]} R_t\right]}{\mathbb{E}_{\pi_{const}^*}\left[\sum_{t \in [T]} R_t\right]} \ \leq \ \frac{B_U}{B_L}. \tag{5}$$

We can interpret the ratio $\frac{B_U}{B_L}$ as a degree of how far the reward function $r$ deviates from the additivity. Then, Theorem 2 implies that the optimal constant policy can be optimal when the reward function is effectively additive. Indeed, when the reward function is additive, i.e., $B_U = B_L$, the exact optimality of optimal constant policy $\pi_{const}^*$ is guaranteed:

**Corollary 1.** *Given an additive reward $r$, $\pi_{const}^*$ is exactly optimal.*

The proof of Theorem 2 is provided in Appendix C.2.

## 4    PROPOSED METHOD: CRUCB

We propose the **C**ombinatorial **R**ising **UCB** (CRUCB) algorithm, presented in Algorithm 1. At each round, CRUCB proceeds in two stages: (i) it estimates the potential of each base arm using *Future-UCB* index based on the recent average outcome, the estimated rate of improvement and an exploration bonus, and then (ii) it calls Solver to select the best super arm after solving a combinatorial optimization problem over the estimated indices.

---
**Algorithm 1** Combinatorial Rising UCB (CRUCB)
---
**Input** $N_{i,0} \leftarrow 0$ for all $i \in [K]$, Sliding window parameter $\varepsilon$.
**Initialize** Play an arbitrary super arm including base arm $i$ twice for each $i \in [K]$.
**for** $t \in (2K + 1, \ldots, T)$ **do**
    Calculate Future-UCB $\acute{\mu}_i(t)$ for each base arm, where $\acute{\mu}_i(t)$ is defined in equation 6.
    $S_t \leftarrow \text{Solver}(\acute{\mu}_1(t), \acute{\mu}_2(t), \cdots, \acute{\mu}_K(t))$.
    Play $S_t$ and observe reward $R_t$.
    Update $\mathcal{F}_t$ and $N_{i,t}$.
**end for**

---

**Estimation** At each time $t$, for each base arm $i$, the Future-UCB index $\acute{\mu}_i(t)$ is estimated as follows to predict the potential of base arm $i$:

$$\acute{\mu}_i(t) := \underbrace{\frac{1}{h_i} \sum_{l=N_{i,t-1}-h_i+1}^{N_{i,t-1}} X_i(l)}_{\text{(i) recent average}} + \underbrace{\frac{1}{h_i} \sum_{l=N_{i,t-1}-h_i+1}^{N_{i,t-1}} (t-l) \frac{X_i(l) - X_i(l-h_i)}{h_i}}_{\text{(ii) predicted upper bound of improvement}}$$

$$+ \underbrace{\sigma \left(t - N_{i,t-1} + h_i - 1\right) \sqrt{\frac{10 \log t^3}{h_i^3}}}_{\text{(iii) exploration bonus}}, \tag{6}$$

where $\sigma$ is the standard deviation and $h_i$ is the size of the sliding window governing a bias-variance trade-off between employing few recent observations (less biased), compared to many past observations (less variance). Specifically, we define the window size adaptively as $h_i = \epsilon N_{i,t-1}$, making it grow proportionally with the number of pulls. This design is crucial for balancing initial agility with long-term statistical stability. The hyperparameter $\epsilon$ tunes this bias-variance trade-off: a smaller $\epsilon$ uses a shorter, more recent history, resulting in a less biased but higher-variance estimate that is more agile in detecting changes. Conversely, a larger $\epsilon$ averages over a longer history, providing a more stable, lower-variance estimate that is slower to adapt to the rising reward dynamics.

The index $\acute{\mu}_i(t)$ consists of three parts:

(i) *recent average*: It is the mean of most recent $h_i$ outcomes from playing base arm $i$, and indicates the expected immediate outcome of playing base arm $i$.

(ii) *predicted upper bound of improvement* : $\frac{X_i(l) - X_i(l-h_i)}{h_i}$ is the estimated slope by finite difference method. Then, by linear extrapolation, $(t-l)\frac{X_i(l) - X_i(l-h_i)}{h_i}$ is an estimate of improvement in the average outcome when playing $i$ for $(t - N_{i,t-1})$ times. By the concavity Assumption 1, the expectation of this term is always optimistic compared to the true value.

(iii) *exploration bonus*: It accounts for uncertainty and encourages exploration of arms that have not been sufficiently often. The exploration bonus used here is intentionally larger than typical bonuses in UCB-based algorithms for stationary bandit settings (Auer et al., 2002), because CRUCB predicts future rewards in a rising setting, where uncertainty is inherently greater.

**Solver** After estimating the potential of each base arm, CRUCB employs `Solver`, which solves a combinatorial optimization problem. `Solver` takes the estimated Future-UCB indices of the base arms $\hat{\boldsymbol{\mu}} = [\acute{\mu}_1(t), \cdots \acute{\mu}_K(t)]$ as input and selects the super arm with the highest expected reward, i.e., $\text{Solver}(\hat{\boldsymbol{\mu}}) = \arg\max_S r(S, \hat{\boldsymbol{\mu}})$. This use of a problem-specific optimization oracle is a standard convention in the combinatorial bandit literature (Chen et al. (2013)). The Solver is an interchangeable component, and its implementation depends on the specific combinatorial structure of the task. For example, in the online shortest path problem, `Solver` can be instantiated as Dijkstra's algorithm (Dijkstra, 1959).

## 5 REGRET ANALYSIS

### 5.1 REGRET UPPER BOUND OF CRUCB

We establish an upper bound on the regret of CRUCB and analyze how it adapts to different levels of problem difficulty. To characterize the difficulty of a CRB instance, we introduce a *cumulative increment* $\Upsilon(M, q) := \sum_{l \in [M-1]} \max_{i \in [K]} \{\gamma_i(l)^q\}$ (Metelli et al., 2022). Intuitively, $\Upsilon(M, q)$ quantifies the difficulty of a CRB instance by measuring the overall outcome growth in expected outcomes. Using $\Upsilon(M, q)$, we establish a regret upper bound for CRUCB as follows:

**Theorem 3.** *Assume that the reward function satisfies Lipschitz assumption:*

$$|r(S, \boldsymbol{\mu}) - r(S, \boldsymbol{\mu}')| \le B \sum_{i \in S} |\mu_i - \mu_i'| , \tag{7}$$

*where $B$ is a Lipschitz constant. Let $\pi_\varepsilon$ be CRUCB with $h_i = \varepsilon N_{i,t}$. Under Assumptions 1&2, for $T > 0$, $q \in [0,1]$, and $\varepsilon \in (0, \frac{1}{2})$, we have the following regret upper bound:*

$$Reg(\pi_\varepsilon, T) \leq \underbrace{\left(2 + \frac{L\pi^2}{3}\right) K + \frac{BKT^q}{1-2\varepsilon}\Upsilon\left((1-2\varepsilon)\frac{LT}{K}, q\right)}_{(i)} + \underbrace{\frac{3K}{\varepsilon}\left((2B\sigma T)^{\frac{2}{3}}(6\log 4T)^{\frac{1}{3}}\right)}_{(ii)}, \quad (8)$$

*where $L := \max_{S\in\mathcal{S}}|S|$ is the maximum size of a super arm.*

Term $(i)$ captures the regret caused by the inherent difficulty of the CRB problem, which is related to the rising nature of expected outcomes and the size of a super arm. First, when outcomes of base arms evolve continuously, i.e., $\Upsilon$ is large, identifying the optimal super arm becomes significantly more difficult, since early observations may not reflect the long-term value of each base arm, making it harder to distinguish the optimal super arm without extensive exploration. Second, when the maximum super arm size $L$ is large, the complexity of accurately estimating the combined reward increases, making it harder to confidently identify the optimal super arm. These challenges are quantified by term $(i)$ via the cumulative increment $\Upsilon$ and $L$, which scales as $O(KT^q\Upsilon(\frac{LT}{K}, q))$. Term $(ii)$ captures the regret due to randomness in observed outcomes and scales as $O(KT^{2/3}(\log T)^{1/3})$.

It is important to note that $q$ is not a hyperparameter of the algorithm but a purely analytical tool used in our proof. The algorithm's implementation is completely independent of $q$. Its sole purpose in our analysis is to provide a single, unified regret bound that holds across a wide spectrum of reward-growth patterns by summarizing the cumulative effect of rising rewards. Thus, our theorem guarantees CRUCB's performance uniformly, while the algorithm itself operates without any knowledge of $q$. The proof of Theorem 3 is provided in Appendix C.3.

The dominant term in the regret bound depends on the difficulty of the instance. When $\Upsilon$ is large, corresponding to more difficult instances, term $(i)$ becomes dominant, potentially leading to linear regret $O(T)$. To characterize the effect of problem difficulty on the regret bound, we present the following corollary, which refines the analysis by assuming an explicit upper bound on the slope $\gamma_i$.

**Corollary 2.** *For a non-increasing function $f$, assume $\gamma_i(n) \leq f(n)$ for each $i \in [K]$ and $n \geq 1$. For $T \geq 0$, $q \in [0,1]$, and $\varepsilon \in (0, 1/2)$, the regret of CRUCB $\pi_\varepsilon$ is bounded as follows:*

$$Reg(\pi_\varepsilon, T) = O\left(\max\left(KT^{\frac{2}{3}}(\log T)^{\frac{1}{3}}, KT^q \int_1^{(1-2\varepsilon)\frac{LT}{K}} f(n)^q dn\right)\right). \quad (9)$$

*In particular, we instantiate the regret upper bound with a set of $f$ with various learning difficulties:*

| | | |
|---|---|---|
| *If $f(n) = \exp(-n)$,* | $Reg(\pi_\varepsilon, T) = O(T^{\frac{2}{3}}\log KT^{\frac{1}{3}})$ . | (10) |
| *If $f(n) = (n+1)^{-c}$ and $c \leq 1$,* | $Reg(\pi_\varepsilon, T) = O(T)$ . | (11) |
| *If $f(n) = (n+1)^{-c}$ and $c > 1$,* | $Reg(\pi_\varepsilon, T) = O\left(\max\left(T^{\frac{2}{3}}\log KT^{\frac{1}{3}}, T^{\frac{1}{c}}\log\frac{LT}{K}\right)\right)$ . | (12) |

To make the role of problem difficulty more explicit, Corollary 2 reformulates the regret bound in terms of $f(n)$. This allows the cumulative increment to be explicitly bounded in terms of $f(n)$, enabling an analytical characterization of the regret.

The regret bounds given in Corollary 2 reflect how the difficulty of the CRB instance varies with the choice of $f(n)$, as illustrated in Figure 2. When $f(n) = \exp(-n)$, outcomes saturated rapidly, making it feasible to disregard the rising nature and resulting in sub-linear regret. In contrast, when $f(n) = (n+1)^{-c}$ with $c \leq 1$, outcomes change gradually, necessitating sustained exploration, and consequently resulting in linear regret. An interesting intermediate regime appears when $f(n) = (n+1)^{-c}$ with $c > 1$, where the regret upper bound explicitly depends on the problem difficulty (parameter $c$), highlighting adaptivity of CRUCB. This adaptivity will become clearer through the regret lower bound analysis in next section with Figure 3.

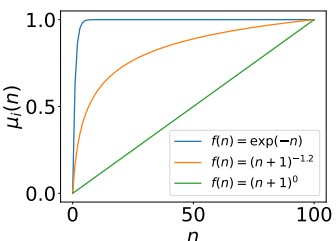

Figure 2: **Growth of outcomes.** $\mu_i(n)$ induced by $\gamma_i(n) = Cf(n)$, with $C$ as a normalizing constant.

## 5.2 REGRET LOWER BOUND OF CRB

In this section, we establish regret lower bounds for CRB. Our results highlight two key findings. First, without any additional assumptions, the regret lower bound is $\Omega(T)$, reflecting the intrinsic difficulty of CRB. Second, given restricted outcome growth, the regret lower bound can be sub-linear. To analyze the regret across a class of CRB instances, we make the dependence on the instance $\nu$ explicit and write the regret as $Reg_\nu(\pi, T)$ in this section.

We begin with a general class of CRB without any structural assumptions on the slope $\gamma_i$.

**Theorem 4.** *(Regret lower bound over a **general** class) Fix sufficiently large time $T$. Let $\mathcal{I}$ be the set of all available CRB instances. Then, any policy $\pi$ suffers regret:*

$$\min_\pi \max_{\nu \in \mathcal{I}} Reg_\nu(\pi, T) = \Omega(LT) , \tag{13}$$

*where $L$ is the maximum size of super arms.*

Theorem 4 establishes that, without any structural assumptions, no algorithm can achieve sub-linear regret. However, as seen in the regret upper bound analysis of CRUCB (Corollary 2), not all instances necessarily incur linear regret. This discrepancy motivates a finer analysis: by considering a more fine-grained instance class, we can distinguish between instances that are inherently difficult and those that allow efficient learning, which the regret lower bound becomes sub-linear. The proof of Theorem 4 is provided in Appendix C.4.

**Theorem 5.** *(Regret lower bound over a **fine-grained** class) Fix sufficiently large time $T$. For an arbitrary constant $1 < c < 2$, define a fine-grained set of CRB instances $\mathcal{A}_c$ as follows:*

$$\mathcal{A}_c := \left\{ \nu : \gamma_i(n) \le (n+1)^{-c}, i \in [K], n \in [T-1] \right\} . \tag{14}$$

*Then, for any policy $\pi$ incurs regret:*

$$\min_\pi \max_{\nu \in \mathcal{A}_c} Reg_\nu(\pi, T) = \Omega\left(\max\left\{ L\sqrt{T}, LT^{2-c} \right\}\right) , \tag{15}$$

*where $L := \max_{S \in \mathcal{S}} |S|$ is the maximum size of a super arm.*

Theorem 5 characterizes how the regret lower bound varies with a parameter $c$. As also reflected in the upper bound, $c$ serves as a structural separator between easy and difficult instances: a larger $c$ leads to slower outcome growth and a smaller regret lower bound, while smaller $c$ result in faster growth and higher regret lower bound. The proof of Theorem 5 is provided in Appendix C.5.

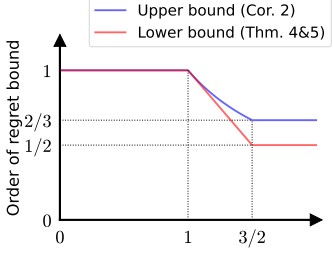

As a final remark for Section 5, our CRUCB achieves a regret upper bound that closely matches the regret lower bound of the CRB (see Figure 3). In particular, without requiring any prior knowledge about the difficulty of the problem instance (e.g., the outcome growth parameter $c$), CRUCB effectively adapts to varying problem difficulties, ensuring robustness of CRUCB across diverse scenarios. To the best of our knowledge, this represents the first explicit and rigorous comparison between regret upper and lower bounds in the rising bandit literature, highlighting a key theoretical contribution of our work.

Figure 3: **Regret bound gap.** The regret lower bound of CRB and the regret upper bound of CRUCB when $f(n) = (n+1)^{-c}$. For $c \le 1$, both the upper and lower bounds are equal to 1. Specifically, for $1 < c < 1.5$, the lower bound $(2-c)$ and the upper bound $\left(\frac{1}{c}\right)$ are of similar order, indicating that the regret bounds closely match.

## 6 EXPERIMENTS

We evaluate the performance of CRUCB against existing state-of-the-art algorithms for rising and non-stationary bandits on the online shortest path planning, in both synthetic environments (Section 6.1) and realistic deep reinforcement learning applications (Section 6.2). Unlike prior works that mainly focus on simplified rising bandit settings, our evaluation further considers realistic deep RL scenarios, underscoring the practical relevance and robustness of CRUCB. Additional results on diverse combinatorial tasks, including maximum weighted matching, minimum spanning tree, and the k-MAX problem are provided in Appendix F.

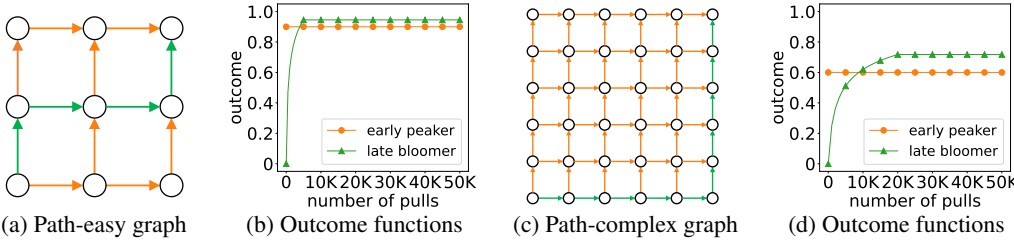

Figure 4: **Online shortest path planning task.** (a, c) Graphs used to evaluate CRUCB and baselines. (b, d) Corresponding outcome functions for each task.

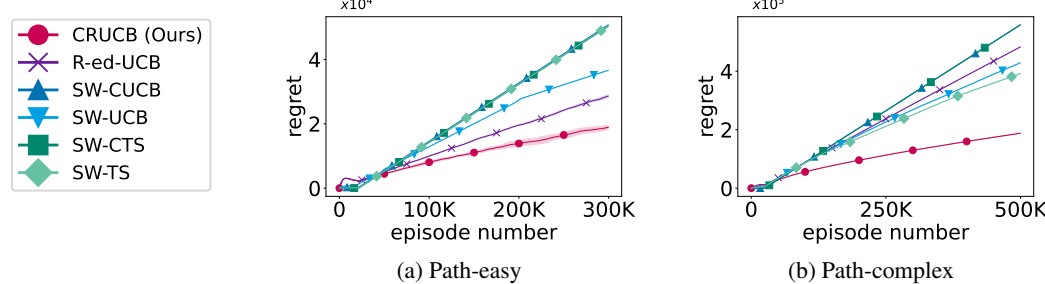

Figure 5: **Cumulative regret in synthetic environments.** Regret curves for (a) Path-easy and (b) Path-complex. Lines show average; shaded areas indicate 99% confidence intervals over 5 runs.

**Baselines** We consider the following baseline algorithms:

- **R-ed-UCB (Metelli et al., 2022)** is a non-combinatorial algorithm for rising bandits, combining a sliding window with UCB-based estimation designed for rising rewards.
- **SW-UCB (Garivier & Moulines, 2011)** and **SW-TS (Trovo et al., 2020)** are non-stationary non-combinatorial bandit algorithms that use a sliding-window approach with UCB and Thompson Sampling, respectively.
- **SW-CUCB (Chen et al., 2021)** and **SW-CTS** are non-stationary combinatorial bandit algorithms that use a sliding-window approach with UCB and Thompson Sampling, respectively.

Detailed pseudocode and descriptions of the baselines are provided in Appendix D. For CRUCB, we set the window size parameter $\epsilon = 0.125$ in our main experiments. We found this to be a robust choice, and a detailed sensitivity analysis on the impact of $\epsilon$ is provided in Appendix F.4.

## 6.1 SYNTHETIC ENVIRONMENTS

We conduct experiments on the online shortest path task using the graph structures shown in Figures 4a and c, each containing two types of edges: *early peakers* and *late bloomers*, as illustrated in Figures 4b and d, respectively. In these experiments, we assume the additive reward setting in which the reward of a super arm is defined as the sum of the outcomes of constituent base arms. In this setting, Corollary 1 implies that the optimal policy is a constant policy repeatedly selecting a fixed path (super arm), which in our experiments corresponds to a path composed solely of late bloomers.

As shown in Figure 5a, CRUCB demonstrates lower regret compared to all baselines in the Path-easy task. R-ed-UCB underperforms despite the simplicity of the graph structure, due to the partially shared enhancement described earlier in Figure 1. In the more complex Path-complex task, CRUCB continues to outperform all baselines, with the gap between CRUCB and R-ed-UCB becomes significantly larger, as shown in Figure 5b.

This is because the effects of the partially shared enhancement are amplified as the overlap of edges (base arms) among paths (super arms) increases. Interestingly, across both environments, non-combinatorial and non-stationary algorithms (SW-UCB, SW-TS) consistently outperform their combinatorial counterparts (SW-CUCB, SW-CTS), with the gap becoming more pronounced in the complex task. This occurs because the increased number of paths promotes broader exploration, allowing non-combinatorial algorithms sufficient time to explore late bloomers, whereas combinatorial algorithms tend to focus exploitation on early peakers, thereby restricting the opportunities for late bloomers to enhance their full reward potential.

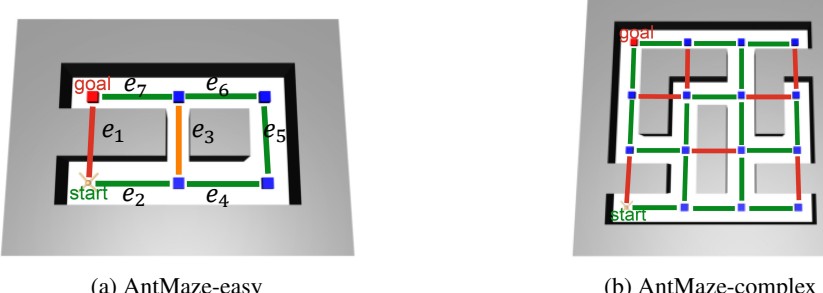

(a) AntMaze-easy  (b) AntMaze-complex

Figure 6: **Deep RL environments.** An ant robot navigates the shortest path from start to goal via intermediate nodes, encountering three types of edges in (a): impossible ($e_1$), bottleneck ($e_3$), and wide ($e_2, e_4, e_5, e_6, e_7$) edge. (b) Focus on impossible and wide edges in a complex map.

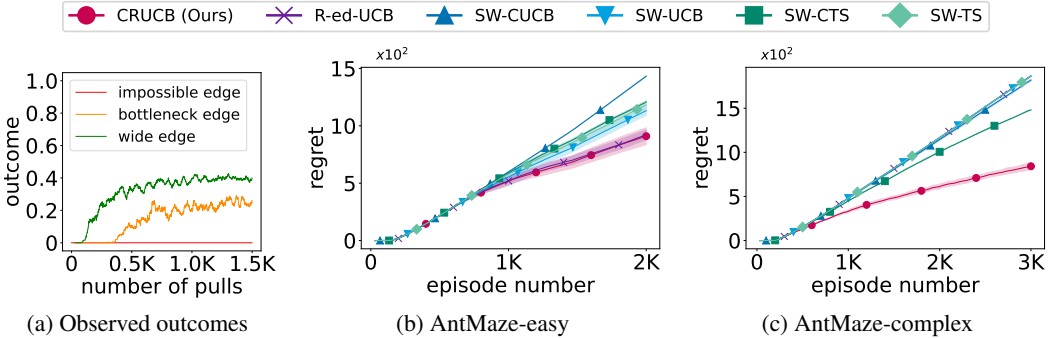

(a) Observed outcomes  (b) AntMaze-easy  (c) AntMaze-complex

Figure 7: **Cumulative regret in deep reinforcement learning environments.** (a) Observed outcomes for each edge with respect to the number of pulls. Regret curves for (b) AntMaze-easy and (c) AntMaze-complex. Lines show average; shaded areas indicate 99% confidence intervals over 5 runs.

## 6.2 DEEP REINFORCEMENT LEARNING

We conduct experiments on the online shortest path problem using hierarchical reinforcement learning in AntMaze environments (Yoon et al., 2024), as illustrated in Figure 6. It divides tasks into high-level and low-level policies. The high-level policy makes abstract decisions, such as the path from start to goal, while the low-level policy executes these decisions by controlling the specific movements of the robot. In our setup, the high-level policy plays a role similar to the CRB framework by selecting paths as super arms, where each edge corresponds to a base arm. As training progresses, the improvements in the low-level policy lead to the rising outcomes for the high-level policy. We consider two tasks: AntMaze-easy and AntMaze-complex (Figures 6a and b). In AntMaze-easy, the policy can choose among three paths: an impossible path using edge ($e_1$), a shortcut path ($e_2, e_3, e_7$) that is short but contains a bottleneck edge $e_3$ requiring more episodes to train, and a detour path ($e_2, e_4, e_5, e_6, e_7$) composed solely of wide edges but requiring more steps. The key challenge in this task is to recognize the rising outcome of the bottleneck edge and efficiently exploit the shortcut path despite its initial difficulty. In AntMaze-complex, the environment has a complex graph structure with extensive paths from start to goal. The large number of paths increases combinatorial complexity, making exploration and identification of the optimal path challenging. Each task aims at a distinct challenge: AntMaze-easy focuses on capturing the rising reward nature, whereas AntMaze-complex emphasizes robustness against growing combinatorial complexity. Detailed descriptions of the environments and reward structures are provided in Appendix E.2.

As depicted in Figure 7a, the outcomes exhibit non-concave behavior due to an extended zero-reward period before the first success; however, the outcome growth appears roughly concave once the rewards increase. Despite this violation of the concavity assumption (Assumption 1), Figures 7b and c show that CRUCB outperforms the baselines, highlighting its robustness in settings where theoretical assumptions are not strictly satisfied. In AntMaze-easy, CRUCB and R-ed-UCB outperform other baselines, as shown in Figure 7b. Given the simplicity of the environment, which includes only three paths, most algorithms successfully identify the detour path. However, non-stationary bandit algorithms tend to exploit the detour path once found and limit further exploration. In contrast, rising bandit algorithms continue to explore the bottleneck path, eventually identifying the optimal path and resulting in lower cumulative regret.

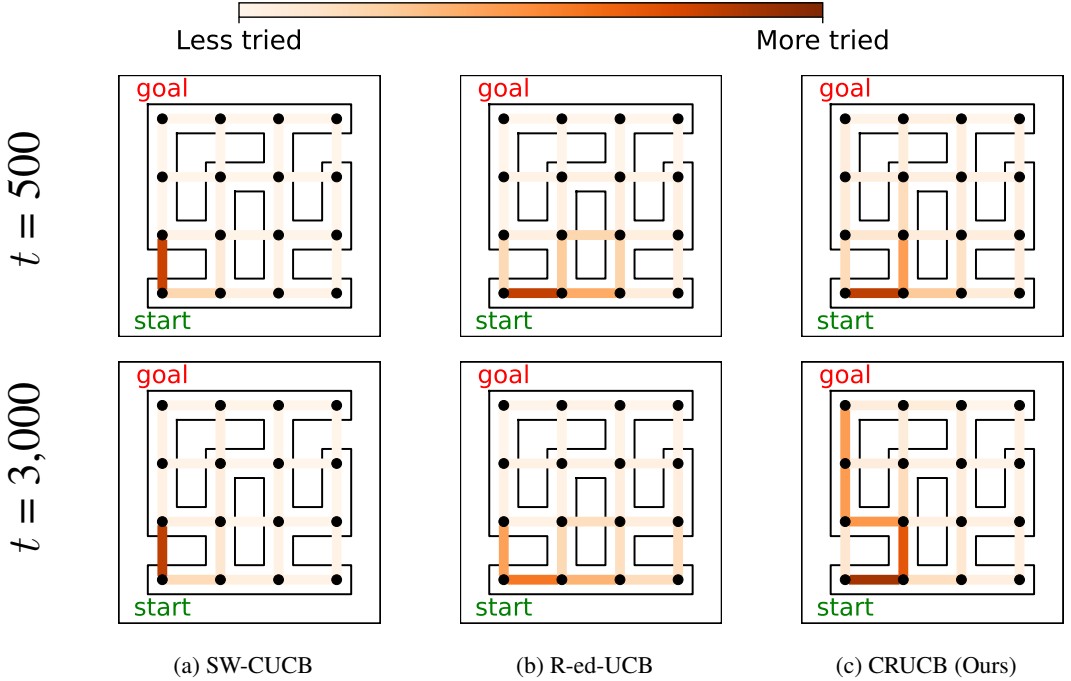

Figure 8: **Heatmap of visit frequencies in AntMaze-complex.** We visualize the visit frequencies of SW-CUCB, R-ed-UCB, and CRUCB at time steps 500 and 3,000 to highlight their respective exploration patterns. Visualizations of other baselines are provided in Appendix G.

As depicted in Figure 8, existing algorithms fail to capture both the combinatorial structure and the rising nature simultaneously. Figure 8a shows thick traces around blocked walls, indicating that the agent repeatedly attempts the same impossible edges. This behavior stems from the agent's evaluation, where it perceives a single impossible edge as more optimistic path than a detour path composed of multiple low-reward edges. Conversely, R-ed-UCB performs uniform exploration as illustrated in Figure 8b. This broad search is an unavoidable consequence of initially treating all 178 possible paths as independent super arms. Even after a sufficient amount of time, its inability to leverage partially shared enhancements leads to incorrect estimations, causing the agent to continue exploring various paths instead of converging on the optimal path. In contrast, CRUCB, as depicted in Figure 8c, integrates both perspectives: it avoids repeated trials on impossible paths, efficiently exploits shared improvements, and quickly concentrates on the optimal path. These observations confirm that the limitations of existing approaches highlighted in Section 1 arise in practice and demonstrate that CRUCB successfully overcomes them.

## 7 CONCLUSION

In this work, we introduced the Combinatorial Rising Bandit (CRB) framework, modeling combinatorial online learning scenarios wherein selecting a super arm enhances the future rewards of its constituent base arms. By highlighting the novel challenges from the *partially shared enhancement* in Figure 1, we established that CRB fundamentally differs from classical bandit formulations.To address this challenge, we developed Combinatorial Rising UCB (CRUCB), a provably efficient algorithm. Our extensive experiments across synthetic and deep RL environments demonstrate that CRUCB robustly handles the combinatorial rising structure where prior methods fail. At the same time, our theoretical analysis establishes tight regret bounds, showing that the algorithm is nearly optimal from an analytical standpoint. Taken together, these results highlight that CRUCB offers both tangible benefits in practice and solid guarantees in theory. While our analysis relies on simplifying assumptions, such as a fixed set of base arms and a static combinatorial structure, these are often reasonable in domains where the action space is pre-defined. However, in certain applications, such as robotic systems that involve skill discovery, the set of feasible actions may evolve over time. Extending CRB to handle such dynamic structures is a promising direction for future research.

ACKNOWLEDGMENTS

This work was supported by the Institute for Information & Communications Technology Planning & Evaluation (IITP) grant funded by the Korean government (MSIT) (No.RS-2019-II191906, Artificial Intelligence Graduate School Program (POSTECH); No.RS-2024-00436680, Global Research Support Program in the Digital Field; No.RS-2024-00457882, AI Research Hub Project); the Korea Institute for Advancement of Technology (KIAT) grant funded by the Ministry of Trade, Industry and Energy (MOTIE) (No.RS-2025-00564342); the Seoul R&BD Program (No.SP240008) through the Seoul Business Agency (SBA) funded by the Seoul Metropolitan Government. This project is supported by Microsoft Research Asia.

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

This material provides proof of theorems, details of environments and baselines, and additional experimental results:

- **Appendix A:** Motivating applications of CRB.
- **Appendix B:** Comparison with existing rising bandit studies.
- **Appendix C:** Proofs of Theorem 1, 2, 3, 4, and 5.
- **Appendix D:** Detailed description and pseudocode of the baselines in Section 6.
- **Appendix E:** Detailed description of the environments in Section 6.
- **Appendix F:** Additional experiments on other combinatorial tasks.
- **Appendix G:** Further analysis of exploration on the deep RL environment.
- **Appendix H:** The use of Large Language Models.

# A    REAL-WORLD APPLICATIONS OF THE CRB FRAMEWORK

The CRB framework, which addresses a regret minimization problem, naturally arises in real-world scenarios where complex actions are composed of reusable sub-actions that improve through repetition. We can consider following applications:

**Network Routing**    optimizes performance metrics such as latency or throughput by selecting network paths (super arms) composed of individual links (base arms). Frequent utilization of specific links enables routing protocols to adapt and improve via better congestion estimation and traffic-pattern learning. Thus, network routing naturally aligns with regret minimization as it balances exploiting known effective routes and exploring potentially better alternatives.

**Crowdsourcing**    aims to find optimal task assignments by combining annotators (base arms) with datasets, which can be framed as an online combinatorial regret minimization problem (Chen et al., 2016). Annotators' skill levels improve over repeated tasks, increasing their annotation accuracy. CRB effectively addresses regret minimization here by dynamically reallocating tasks among annotators to leverage their rising skills, thereby optimizing overall annotation quality and cost-effectiveness.

**LLM-based Planning**    decomposes complex tasks into simpler subtasks (base arms), akin to the Chain-of-Thought (CoT) approach (Wei et al., 2022). Iterative prompting, where previous outputs refine future ones (Zheng et al., 2023), enhances model performance over time (rising reward). By applying CRB to decompose tasks into subtasks, we expect improved performance by exploiting the rising reward structure in these iterative reasoning tasks.

## B   COMPARISON WITH EXISTING RISING BANDIT STUDIES

The rising bandit problem has been widely studied in non-combinatorial settings (Fiandri et al., 2024a;b; Heidari et al., 2016; Metelli et al., 2022; Mussi et al., 2024; Patil et al., 2022; Xia et al., 2024; Amichay & Mansour, 2025), where each base arm evolves independently over pulls. In this work, we consider a combinatorial extension of the rising bandit problem, where each action is a set of base arms. This generalization introduces new challenges that fundamentally differ from previous work.

In previous work (Heidari et al., 2016), it is shown that a constant policy is optimal in the rising bandit setting. However in Section 3, we demonstrate that in the combinatorial setting, constant policies are generally not optimal. Furthermore, (Metelli et al., 2022) focus primarily on establishing worst-case regret lower bounds, showing that regret is linear ($\Omega(T)$), highlighting the inherent difficulty of the problem. In contrast, we show that under a more fine-grained instance class where reward growth is bounded, the regret lower bound can be sublinear. Moreover, we illustrate that this lower bound is tight, nearly matching it with the regret upper bound of our proposed algorithm, CRUCB.

A recent study (Genalti et al., 2024) investigates rising bandits with structured dependencies among arms, introducing a graph-triggered mechanism in which pulling an arm increases the rewards of its neighboring arms. While conceptually related to our work, their approach assumes uniform enhancement across neighbors, without modeling the nuanced structure of overlapping actions. In contrast, our CRB framework models partially shared enhancement, preserving the combinatorial structure. This distinction makes CRB a more general and unified framework for capturing rising reward dynamics in combinatorial settings.

## C  PROOF OF THEOREMS

### C.1  PROOF OF THEOREM 1

It suffices to show that there exists a CRB instance such that the *constant policy* is not the optimal policy. We consider $k$-MAX problem, where reward function is given as follows:

$$r(S, \boldsymbol{\mu}) := \max_{i \in S}(\mu_i) . \tag{16}$$

Note that equation 16 satisfies all assumptions. Consider $\boldsymbol{\mu}$ such that:

$$\mu_1(n) = \begin{cases} \frac{10}{T}n & n < \frac{T}{10} \\ 1 & n \geq \frac{T}{10} \end{cases} , \tag{17}$$

$$\mu_2(n) = \begin{cases} 0.1 & n = 1 \\ 0.9 & n > 1 \end{cases} , \tag{18}$$

$$\mu_3(n) = 0.5 . \tag{19}$$

Let $K = 3$, $\mathcal{S} = \{(1,2), (1,3), (2,3)\}$, $T \gg 100$. For simplicity, when a base arm is pulled for $n$-th times, then the outcome is $\mu_i(n)$ without considering randomness. In this problem instance, the optimal constant policy is selecting the super arm $(1,2)$ continuously. For the best constant policy, it receives $0.1$ for $t = 1$, and $\frac{10t}{T}$ for $1 < t \leq \frac{9T}{10}$ and $1$ for $t > \frac{9T}{10}$. However, if $(2,3)$ is firstly selected once and $(1,2)$ for the remaining time, it receives $0.3$ more rewards than the best constant policy. This is because selecting $(2,3)$ initially yields an immediate gain of $0.4$ from first selecting, but later results in a loss of $0.1$ due to not playing optimal super arm $(1,2)$. Consequently, the total reward is higher than that of the best constant policy, which suffices to complete proof.

### C.2  PROOF OF THEOREM 2

For proof, we first consider a specific case: additive reward.

**Lemma 1.** *Given an additive reward $r(S, \boldsymbol{\mu}) = \sum_{i \in S} \mu_i$, $\pi^*_{const}$ is exactly optimal.*

**Set up.**  Since we consider additive reward function, the cumulative reward is invariant with respect to permutations of the order of selecting super arms, which means that a policy can be represented as the vector of number of pulling each super arm, that is, a policy $\pi$ can be represented as follows:

$$\pi \mapsto \left( T_1^\pi, T_2^\pi, \cdots, T_{|\mathcal{S}|}^\pi \right) , \tag{20}$$

where $T_S^\pi$ denotes the number of pulling a super arm $S$ until time $T$ by the policy $\pi$, which satisfies $\sum_{S \in \mathcal{S}} T_S^\pi = T$. Let $N_{i,T}^\pi$ denote the number of selecting a base arm $i$ until time $T$ by $\pi$. Then, $N_{i,T}^\pi$ can be represented as follows:

$$N_{i,T}^\pi = \sum_{S \in \mathcal{S}} T_S^\pi \mathbb{1} \{i \in S\} , \tag{21}$$

where $\mathbb{1}$ denotes the indicator function. Let $\pi^*$ be the optimal policy given $\boldsymbol{\mu}$ and $T$. We show that if $\pi^*$ pulls at least two different super arms, then a constant policy can be constructed so that generates larger than or equal to the expected cumulative reward as the one produced by $\pi^*$, which suffices to conclude.

Assume that $\pi^*$ selects $m$ distinct super arms, denoted by super arms as $S_1, S_2, \ldots, S_m$. Define a subset of base arms $B_c$ and $B_j$ for each $j \in [m]$ as follows:

$$B_c := \{i \in [K] : i \in S_j, \quad \forall j \in [m]\} , \tag{22}$$

$$B_j := S_j \setminus B_c . \tag{23}$$

$B_c$ represents the subset of the common base arms included in every selected super arm by the optimal policy $\pi^*$ and $B_j$ represents the subset of base arms included in the super arm $S_j$ except for $B_c$.

**Claim 1.** $\sum_{i \in B_j} \mu_i(N_{i,T}^{\pi^*})$ is equal for all $j \in [m]$.

*Proof.* To establish Claim 1, we consider two arbitrary distinct super arms $S_1$ and $S_2$, without loss of generality. We observe $\sum_{i \in B_1 \setminus B_2} \mu_i(N_{i,T}^{\pi^*}) \geq \sum_{i \in B_2 \setminus B_1} \mu_i(N_{i,T}^{\pi^*})$. If not, that is, $\sum_{i \in B_1 \setminus B_2} \mu_i(N_{i,T}^{\pi^*}) < \sum_{i \in B_2 \setminus B_1} \mu_i(N_{i,T}^{\pi^*})$, we can construct new policy $\pi_1$ as follows:

$$T_S^{\pi_1} = \begin{cases} T_{S_1}^{\pi^*} - 1 & S = S_1 \\ T_{S_2}^{\pi^*} + 1 & S = S_2 \\ T_S^{\pi^*} & \text{otherwise.} \end{cases} \tag{24}$$

Then, $N_{i,T}^{\pi_1}$ is given by:

$$N_{i,T}^{\pi_1} = \begin{cases} N_{i,T}^{\pi^*} - 1 & i \in B_1 \setminus B_2 \\ N_{i,T}^{\pi^*} + 1 & i \in B_2 \setminus B_1 \\ N_{i,T}^{\pi^*} & \text{otherwise .} \end{cases} \tag{25}$$

The difference between the expected cumulative reward of $\pi^*$ and $\pi_1$ is given by:

$$\sum_{i \in [K]} \left( \sum_{n \in [N_{i,T}^{\pi^*}]} \mu_i(n) - \sum_{n \in [N_{i,T}^{\pi_1}]} \mu_i(n) \right) \tag{26}$$

$$= \sum_{i \in B_1 \setminus B_2} \mu_i(N_{i,T}^{\pi^*}) - \sum_{i \in B_2 \setminus B_1} \mu_i(N_{i,T}^{\pi^*} + 1) \tag{27}$$

$$< 0 , \tag{28}$$

which indicates that the cumulative reward of $\pi_1$ is larger than that of $\pi^*$. However, it is contradicting with the assumption that $\pi^*$ is optimal and thus we have $\sum_{i \in B_1 \setminus B_2} \mu_i(N_{i,T}^{\pi^*}) \geq \sum_{i \in B_2 \setminus B_1} \mu_i(N_{i,T}^{\pi^*})$. By applying the same logic, we can also derive that $\sum_{i \in B_1 \setminus B_2} \mu_i(N_{i,T}^{\pi^*}) \leq \sum_{i \in B_2 \setminus B_1} \mu_i(N_{i,T}^{\pi^*})$. Combing these results, we have $\sum_{i \in B_1 \setminus B_2} \mu_i(N_{i,T}^{\pi^*}) = \sum_{i \in B_2 \setminus B_1} \mu_i(N_{i,T}^{\pi^*})$. By adding $\sum_{i \in B_1 \cap B_2} \mu_i(N_{i,T}^{\pi^*})$, we can derive that $\sum_{i \in B_1} \mu_i(N_{i,T}^{\pi^*}) = \sum_{i \in B_2} \mu_i(N_{i,T}^{\pi^*})$. Since we can apply the same logic to any arbitrary super arm pair, we conclude the claim. □

**Claim 2.** $\sum_{i \in B_j \setminus B_{j'}} \mu_i(N_{i,T}^{\pi^*} - T_{S_1}^{\pi^*} + 1) = \sum_{i \in B_j \setminus B_{j'}} \mu_i(N_{i,T}^{\pi^*})$ for any $j, j' \in [m]$ .

*Proof.* Similar to Claim 1, we consider $S_1$ and $S_2$, without loss of generality. Given that $\sum_{i \in B_1 \setminus B_2} \mu_i(N_{i,T}^{\pi^*}) \leq \sum_{i \in B_2 \setminus B_1} \mu_i(N_{i,T}^{\pi^*})$ from preceding analysis, we observe $\sum_{i \in B_1 \setminus B_2} \mu_i(N_{i,T}^{\pi^*} - T_{S_1}^{\pi^*} + 1) \geq \sum_{i \in B_2 \setminus B_1} \mu_i(N_{i,T}^{\pi^*})$. Otherwise, that is, $\sum_{i \in B_1 \setminus B_2} \mu_i(N_{i,T}^{\pi^*} - T_{S_1}^{\pi^*} + 1) < \sum_{i \in B_2 \setminus B_1} \mu_i(N_{i,T}^{\pi^*})$, we can construct new policy $\pi_2$ such that:

$$T_S^{\pi_2} = \begin{cases} 0 & S = S_1 \\ T_{S_1}^{\pi^*} + T_{S_2}^{\pi^*} & S = S_2 \\ T_S^{\pi^*} & \text{otherwise.} \end{cases} \tag{29}$$

Then, $N_{i,T}^{\pi_2}$ is given by:

$$N_{i,T}^{\pi_2} = \begin{cases} N_{i,T}^{\pi^*} - T_{S_1}^{\pi^*} & i \in B_1 \setminus B_2 \\ N_{i,T}^{\pi^*} + T_{S_1}^{\pi^*} & i \in B_2 \setminus B_1 \\ N_{i,T}^{\pi^*} & \text{otherwise .} \end{cases} \tag{30}$$

The difference between the cumulative rewards of $\pi^*$ and $\pi_2$ is given by:

$$\sum_{i \in [K]} \left( \sum_{n \in [N_{i,T}^{\pi^*}]} \mu_i(n) - \sum_{n \in [N_{i,T}^{\pi_2}]} \mu_i(n) \right) \tag{31}$$

$$= \sum_{i \in B_1 \setminus B_2} \sum_{n = N_{i,T}^{\pi^*} - T_{S_1}^{\pi^*} + 1}^{N_{i,T}^{\pi^*}} \mu_i(n) - \sum_{i \in B_2 \setminus B_1} \sum_{n = N_{i,T}^{\pi^*} + 1}^{N_{i,T}^{\pi^*} + T_{S_1}^{\pi^*}} \mu_i(n) \tag{32}$$

$$= \sum_{l \in [T_{S_1}^{\pi^*}]} \left( \sum_{i \in B_1 \setminus B_2} \mu_i(N_{i,T}^{\pi^*} - T_{S_1}^{\pi^*} + l) - \sum_{i \in B_2 \setminus B_1} \mu_i(N_{i,T}^{\pi^*} + l) \right) \tag{33}$$

$$\leq \left( \sum_{i \in B_1 \setminus B_2} \mu_i(N_{i,T}^{\pi^*} - T_{S_1}^{\pi^*} + 1) - \sum_{i \in B_2 \setminus B_1} \mu_i(N_{i,T}^{\pi^*}) \right) \tag{34}$$

$$+ (T_{S_1}^{\pi^*} - 1) \left( \sum_{i \in B_1 \setminus B_2} \mu_i(N_{i,T}^{\pi^*}) - \sum_{i \in B_2 \setminus B_1} \mu_i(N_{i,T}^{\pi^*}) \right)$$

$$< 0 , \tag{35}$$

where equation 34 holds since $\mu_i(N_{i,T}^{\pi^*} - T_{S_1}^{\pi^*} + l) \leq \mu_i(N_{i,T}^{\pi^*})$ and $\mu_i(N_{i,T}^{\pi^*} + l) > \mu_i(N_{i,T}^{\pi^*})$ for any $l \in [2, T_{S_1}^{\pi^*}]$ for any base arm $i$ by the definition of combinatorial rising bandit. It indicates that the cumulative reward of $\pi_2$ is larger than that of $\pi^*$, which is a contradiction with assumption that $\pi^*$ is optimal. Therefore, we have $\sum_{i \in B_1 \setminus B_2} \mu_i(N_{i,T}^{\pi^*} - T_{S_1}^{\pi^*} + 1) \geq \sum_{i \in B_2 \setminus B_1} \mu_i(N_{i,T}^{\pi^*})$. Combining this observation with the previous observation, we have $\sum_{i \in B_1 \setminus B_2} \mu_i(N_{i,T}^{\pi^*} - T_{S_1}^{\pi^*} + 1) = \sum_{i \in B_1 \setminus B_2} \mu_i(N_{i,T}^{\pi^*})$. This result implies that rewards of all base arms in $S_1$ are flat after pulling for $N_{i,T}^{\pi^*} - T_{S_1}^{\pi^*}$ times. Since we can apply the same logic to any arbitrary super arm pair, we conclude the claim. $\square$

**Induction.** Lastly, we construct constant policy inductively. As before, we choose two arbitrary two super arm and consider $S_1$ and $S_2$ without loss of generality. we revisit $\pi_2$. By Claim 1 and Claim 2 the difference between $\pi^*$ and $\pi_2$ equals 0, which means that $\pi_2$ is also an optimal policy. We remark that $\pi_2$ plays $m - 1$ distinct super arms. Applying preceding logic inductively, we can construct the optimal policy pulls only one super arm, which completes proof for Lemma 1.

Then, we are ready to prove Theorem 2.

*Proof.* For the proof, we define $S'_{\text{const}}$ and $\pi'_{\text{const}}$ as follows:

$$S'_{\text{const}} := \arg\max_S \sum_{t \in [T]} \sum_{i \in S} \mu_i(t - 1) , \tag{36}$$

$$S^*_{\text{const}} := \arg\max_S \sum_{t \in [T]} r(S, \boldsymbol{\mu}_S(t - 1)) , \tag{37}$$

$$\pi'_{\text{const}}(t) := S'_{const} \quad \forall t \in [T] , \tag{38}$$

$$\pi^*_{\text{const}}(t) := S^*_{const} \quad \forall t \in [T] , \tag{39}$$

where $\boldsymbol{\mu}_S(t - 1) := \{\mu_i(t - 1) : i \in S\}$. Intuitively, $\pi'_{\text{const}}$ indicates the optimal constant policy when the reward function is additive and $\pi^*_{\text{const}}$ indicates the optimal constant policy when the reward function is given $r(\cdot)$.

Let $\pi^*$ be optimal policy, and denote the selected super arm and expectation of base arm at time $t$ under $\pi^*$ as $S_t^*$ and $\boldsymbol{\mu}_t^*$ respectively. Then, we have:

$$\mathbb{E}_{\pi^*}\left[\sum_{t\in[T]} R_t\right] = \sum_{t\in[T]} r(S_t^*, \boldsymbol{\mu}_{t-1}^*) \tag{40}$$

$$= \sum_{t\in[T]} \left(r(S_t^*, \boldsymbol{\mu}_{t-1}^*) - r(S_t^*, \mathbf{0}) + r(S_t^*, \mathbf{0})\right) \tag{41}$$

$$\leq B_U \sum_{t\in[T]} \sum_{i\in S_t^*} \mu_i^*(t-1) \tag{42}$$

$$\leq B_U \sum_{t\in[T]} \sum_{i\in S'_{\text{const}}} \mu_i(t-1) . \tag{43}$$

From Lemma 1, we know that the reward under the optimal policy is bounded by the reward under a constant arm selection, which leads to the inequality in equation 43.

Now, consider the reward of $\pi'_{\text{const}}$. Then, we have:

$$\mathbb{E}_{\pi'_{\text{const}}}\left[\sum_{t\in[T]} R_t\right] = \sum_{t\in[T]} r(S'_{\text{const}}, \boldsymbol{\mu}_{S'_{\text{const}}}(t-1)) \tag{44}$$

$$= \sum_{t\in[T]} \left(r(S'_{\text{const}}, \boldsymbol{\mu}_{S'_{\text{const}}}(t-1)) - r(S'_{\text{const}}, \mathbf{0}) + r(S'_{\text{const}}, \mathbf{0})\right) \tag{45}$$

$$\geq B_L \sum_{t\in[T]} \sum_{i\in S'_{\text{const}}} \mu_i(t-1) . \tag{46}$$

From the inequalities equation 43 and equation 46, we can derive the following ratio:

$$\frac{\mathbb{E}_{\pi^*}\left[\sum_{t\in[T]} R_t\right]}{\mathbb{E}_{\pi^*_{\text{const}}}\left[\sum_{t\in[T]} R_t\right]} \leq \frac{B_U \sum_{t\in[T]} \sum_{i\in S'_{\text{const}}} \mu_i(t-1)}{B_L \sum_{t\in[T]} \sum_{i\in S'_{\text{const}}} \mu_i(t-1)} \tag{47}$$

$$= \frac{B_U}{B_L} . \tag{48}$$

Since $S^*_{\text{const}}$ is defined to maximize the reward we have the inequality:

$$\mathbb{E}_{\pi^*_{\text{const}}}\left[\sum_{t\in[T]} R_t\right] > \mathbb{E}_{\pi'_{\text{const}}}\left[\sum_{t\in[T]} R_t\right] . \tag{49}$$

Finally, combining all the inequalities, we conclude:

$$\frac{\mathbb{E}_{\pi^*}\left[\sum_{t\in[T]} R_t\right]}{\mathbb{E}_{\pi^*_{\text{const}}}\left[\sum_{t\in[T]} R_t\right]} \leq \frac{\mathbb{E}_{\pi^*}\left[\sum_{t\in[T]} R_t\right]}{\mathbb{E}_{\pi'_{\text{const}}}\left[\sum_{t\in[T]} R_t\right]} \leq \frac{B_U}{B_L} . \tag{50}$$

$\square$

### C.3 Proof of Theorem 3

*Proof.* We rewrite the regret as follows.

$$Reg(\pi, T) = \sum_{t \in [T]} \mathbb{E}_{\pi^*}[R_t] - \sum_{t \in [T]} \mathbb{E}_\pi[R_t] \tag{51}$$

$$= \sum_{t \in [T]} r(S_t^{\pi^*}, \boldsymbol{\mu}_{S_t^{\pi^*}}) - \sum_{t \in [T]} \mathbb{E}_\pi\left[r(S_t^\pi, \boldsymbol{\mu}_{S_t^\pi})\right] \tag{52}$$

$$= \sum_{t \in [T]} \mathbb{E}_\pi\left[r(S_t^{\pi^*}, \boldsymbol{\mu}_{S_t^{\pi^*}}) - r(S_t^\pi, \boldsymbol{\mu}_{S_t^\pi})\right] , \tag{53}$$

where equation 52 holds since in semi-bandit feedback setting, the optimal policy is characterized as a deterministic policy.

To define well-estimated event, we define $\hat{\mu}_i(t)$ and $\widetilde{\mu}_i(t)$ as follows:

$$\hat{\mu}_i(t) := \frac{1}{h_i} \sum_{l=N_{i,t-1}-h_i+1}^{N_{i,t-1}} \left( X_i(l) + (t-l)\frac{X_i(l) - X_i(l-h_i)}{h_i} \right) \tag{54}$$

$$\widetilde{\mu}_i(t) := \frac{1}{h_i} \sum_{l=N_{i,t-1}-h_i+1}^{N_{i,t-1}} \left( \mu_i(l) + (t-l)\frac{\mu_i(l) - \mu_i(l-h_i)}{h_i} \right) \tag{55}$$

$$\beta_i(t) := \sigma\left(t - N_{i,t-1} + h_i - 1\right) \sqrt{\frac{10 \log t^3}{h_i^3}} \tag{56}$$

$$\acute{\mu}_i(t) := \hat{\mu}_i(t) + \beta_i(t) . \tag{57}$$

We define well-estimated event $\mathcal{E}_t$ as follows:

$$\mathcal{E}_{i,t} := \{|\hat{\mu}_i(t) - \widetilde{\mu}_i(t)| \le \beta_i(t)\} , \tag{58}$$

$$\mathcal{E}_t := \cap_{i \in [K]} \mathcal{E}_{i,t} . \tag{59}$$

We decompose the regret with well-estimated event $\mathcal{E}_t$ as follows:

$$\mathbb{E}_\pi\left[r(S_t^{\pi^*}, \boldsymbol{\mu}_{S_t^{\pi^*}}) - r(S_t^\pi, \boldsymbol{\mu}_{S_t^\pi})\right] \tag{60}$$

$$= \underbrace{\mathbb{E}_\pi\left[\left(r(S_t^{\pi^*}, \boldsymbol{\mu}_{S_t^{\pi^*}}) - r(S_t^\pi, \boldsymbol{\mu}_{S_t^\pi})\right)\mathbb{1}\{\neg\mathcal{E}_t\}\right]}_{(A)} + \underbrace{\mathbb{E}_\pi\left[\left(r(S_t^{\pi^*}, \boldsymbol{\mu}_{S_t^{\pi^*}}) - r(S_t^\pi, \boldsymbol{\mu}_{S_t^\pi})\right)\mathbb{1}\{\mathcal{E}_t\}\right]}_{(B)} .$$

$$\tag{61}$$

Firstly, we bound term (A):

$$\mathbb{E}_\pi\left[\left(r(S_t^{\pi^*}, \boldsymbol{\mu}_{S_t^{\pi^*}}) - r(S_t^\pi, \boldsymbol{\mu}_{S_t^\pi})\right)\mathbb{1}\{\neg\mathcal{E}_t\}\right] \le L \sum_{t \in [T]} \mathbb{E}_\pi\left[\mathbb{1}\{\neg\mathcal{E}_t\}\right] \tag{62}$$

$$= L \sum_{t \in [T]} \mathbb{P}(\neg\mathcal{E}_t) \tag{63}$$

$$\le \sum_{t \in [T]} \frac{2LK}{t^2} \tag{64}$$

$$\le \frac{LK\pi^2}{3} . \tag{65}$$

where equation 64 holds by Lemma 2 and equation 65 holds since $\sum_{t=1}^\infty \frac{1}{t^2} = \frac{\pi^2}{6}$.

**Lemma 2.** *(Metelli et al., 2022) For every round $K < t < T$, and window size $1 \le h_i \le \varepsilon N_{i,t-1}$, we have:*

$$\mathbb{P}(\neg\mathcal{E}_t) \le \frac{2K}{t^2} . \tag{66}$$

Next, we bound term (B). Firstly, we utilize Lipschitz continuity.

$$r(S_t^{\pi^*}, \boldsymbol{\mu}_{S_t^{\pi^*}}) - r(S_t^{\pi}, \boldsymbol{\mu}_{S_t^{\pi}}) \tag{67}$$

$$= r(S_t^{\pi^*}, \boldsymbol{\mu}_{S_t^{\pi^*}}) - r(S_t^{\pi}, \boldsymbol{\mu}_{S_t^{\pi}}) + r(S_t^{\pi}, \boldsymbol{\hat{\mu}}_{S_t^{\pi}}) - r(S_t^{\pi}, \boldsymbol{\hat{\mu}}_{S_t^{\pi}}) + r(S_t^{\pi^*}, \boldsymbol{\hat{\mu}}_{S_t^{\pi^*}}) - r(S_t^{\pi^*}, \boldsymbol{\hat{\mu}}_{S_t^{\pi^*}}) \tag{68}$$

$$\leq -r(S_t^{\pi}, \boldsymbol{\mu}_{S_t^{\pi}}) + r(S_t^{\pi}, \boldsymbol{\hat{\mu}}_{S_t^{\pi}}) - r(S_t^{\pi}, \boldsymbol{\hat{\mu}}_{S_t^{\pi}}) + r(S_t^{\pi^*}, \boldsymbol{\hat{\mu}}_{S_t^{\pi^*}}) \tag{69}$$

$$\leq r(S_t^{\pi}, \boldsymbol{\hat{\mu}}_{S_t^{\pi}}) - r(S_t^{\pi}, \boldsymbol{\mu}_{S_t^{\pi}}) \tag{70}$$

$$\leq B \sum_{i \in S_t^{\pi}} |\hat{\mu}_i(t) - \mu_i(t)| \tag{71}$$

$$\leq \underbrace{B \sum_{i \in S_t^{\pi}} \widetilde{\mu}_i(t) - \mu_i(t)}_{(B1)} + \underbrace{2B \sum_{i \in S_t^{\pi}} \beta_i(t)}_{(B2)}, \tag{72}$$

where equation 69 holds by Assumption 2 and equation 70 holds by definition of CRUCB and equation 71 holds by Lipschitz assumption and equation 72 holds by well-estimated event $\mathcal{E}_t$.

We bound the term (B1) defining $t_{i,n}$ as the time step where the base arm $i$ is pulled the $n^{\text{th}}$ time:

$$\sum_{t \in [T]} \sum_{i \in S_t^{\pi}} \widetilde{\mu}_i(t) - \mu_i(t) \leq 2K + \sum_{i \in [K]} \sum_{n=3}^{N_{i,t}} \min\{\widetilde{\mu}_i(t_{i,n}) - \mu_i(t), 1\} \tag{73}$$

$$\leq 2K + \sum_{i \in [K]} \sum_{n=3}^{N_{i,t}} \min\left\{\frac{1}{2}(2t_{i,n} - 2n + h_i - 1)\gamma_i(n - 2h_i + 1), 1\right\} \tag{74}$$

$$= 2K + \sum_{i \in [K]} \sum_{n=3}^{N_{i,t}} \min\{T\gamma_i((1 - 2\varepsilon)n), 1\} \tag{75}$$

$$\leq 2K + T^q \sum_{i \in [K]} \sum_{n=3}^{N_{i,t}} \gamma_i((1 - 2\varepsilon)n)^q \tag{76}$$

$$\leq 2K + KT^q \left(\frac{1}{1 - 2\varepsilon}\right) \Upsilon\left((1 - 2\varepsilon)\frac{LT}{K}, q\right), \tag{77}$$

where equation 74 follows from the Lemma A.3, in (Metelli et al., 2022), equation 76 follows from the fact $\min(s, 1) \leq \min(s, 1)^q \leq s^q$ for $q \in [0, 1]$, and equation 77 follows from the Lemma C.2. in (Metelli et al., 2022). Now, we bound the term (B2).

$$\sum_{t \in [T]} \sum_{i \in S_t^{\pi}} 2B \min\{\beta_i(t), 1\} = \sum_{t \in [T]} \sum_{i \in S_t} 2B \min\left\{\sigma(t - N_{i,t-1} + h_i - 1)\sqrt{\frac{2\log 4t^3}{h^3}}, 1\right\} \tag{78}$$

$$\leq \sum_{t \in [T]} \sum_{i \in S_t^{\pi}} 2B \min\left\{T\sigma\sqrt{\frac{6\log 4T}{(\varepsilon \lfloor N_{i,t} \rfloor)^3}}, 1\right\} \tag{79}$$

$$= \sum_{i \in [K]} \sum_{n \in [N_{i,t}]} 2B \min\left\{T\sigma\sqrt{\frac{6\log 4T}{(\varepsilon \lfloor n \rfloor)^3}}, 1\right\}. \tag{80}$$

Choose $n' = \frac{(2B\sigma T)^{\frac{2}{3}}(6\log(4T))^{\frac{1}{3}}}{\varepsilon}$. Then for $n > n'$

$$2B\sigma T\sqrt{\frac{6\log 4T}{(\varepsilon \lfloor n \rfloor)^3}} \leq 1. \tag{81}$$

Thus, we have:

$$\sum_{i \in [K]} \sum_{n=1}^{N_{i,t}} 2B \min \left\{ \sigma T \sqrt{\frac{6 \log 4T}{(\varepsilon \lfloor n \rfloor)^3}}, 1 \right\} \leq \sum_{i \in [K]} \left( n' + \sum_{n=n'+1}^{T} 2B\sigma T \sqrt{\frac{6 \log 4T}{(\varepsilon \lfloor n \rfloor)^3}} \right) \quad (82)$$

$$\leq K \left( n' + 2B\sigma T \sqrt{\frac{6 \log 4T}{\varepsilon^3}} \int_{n'}^{\infty} x^{-\frac{3}{2}} dx \right) \quad (83)$$

$$\leq \frac{3K}{\epsilon} \left( (2B\sigma T)^{\frac{2}{3}} (6 \log 4T)^{\frac{1}{3}} \right) , \quad (84)$$

where equation 83 comes from the fact that the sum of monotone decreasing function can be upper bounded. Combining the results from equation 65, equation 77, and equation 84, we conclude the proof. □

### C.4 PROOF OF THEOREM 4

*Proof.* Firstly, we consider non-combinatorial case, which means that every super arm has only one base arm. We construct two different problems and show that no policy can achieve sub-linear regret.

**Lemma 3.** *Let $\mathcal{I}'$ be the set of all available two-armed rising bandit problem. For sufficiently large time $T$, any policy $\pi$ suffers regret:*

$$\min_{\pi} \max_{\boldsymbol{\mu} \in \mathcal{I}'} Reg_{\boldsymbol{\mu}}(\pi, T) \geq \frac{T}{16} , \quad (85)$$

*Proof.* For simplicity, we consider the deterministic problem, that is, $\sigma = 0$. Let $Rew_{\boldsymbol{\mu}}(\pi, T)$ be the cumulative reward of policy $\pi$ up to time $T$ with respect to the problem instance $\boldsymbol{\mu}$. Define two problem $\boldsymbol{\mu^A}$ and $\boldsymbol{\mu^B}$ as follows:

$$\mu_1^A(n) = \mu_1^B(n) = \frac{1}{2}$$

$$\mu_2^A(n) = \begin{cases} \frac{3n}{2T} & \text{if } n \leq \frac{2T}{3} \\ 1 & \text{otherwise} \end{cases}$$

$$\mu_2^B(n) = \begin{cases} \frac{3n}{2T} & \text{if } n \leq \frac{T}{3} \\ \frac{1}{2} & \text{otherwise} \end{cases} .$$

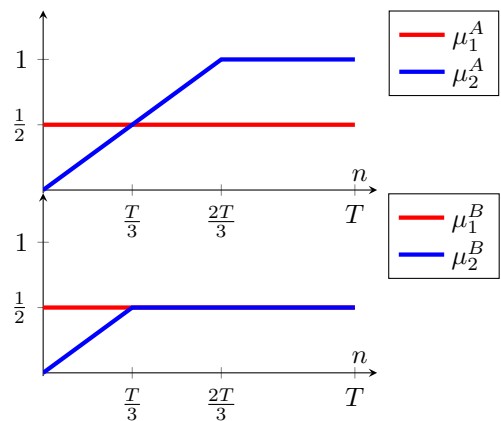

In this setting, we define $\mathcal{S}$ as follows:

$$\mathcal{S} = \{S_1, S_2\} . \quad (86)$$

The main idea of the proof is that for any arbitrary policy $\pi'$, the policy receives the same rewards for both $\boldsymbol{\mu^A}$ and $\boldsymbol{\mu^B}$ at least until $\frac{T}{3}$, indicating that:

$$Rew_{\boldsymbol{\mu^A}} \left( \pi', \frac{T}{3} \right) = Rew_{\boldsymbol{\mu^B}} \left( \pi', \frac{T}{3} \right) . \quad (87)$$

Fix some arbitrary policy $\pi$ and define $M$ as follows:

$$M := \mathbb{E}_{\boldsymbol{\mu}^A, \pi}[N_{S_1, \frac{T}{3}}] = \mathbb{E}_{\boldsymbol{\mu}^B, \pi}[N_{S_1, \frac{T}{3}}] \tag{88}$$

We compute the cumulative regret of policy $\pi$ in $\boldsymbol{\mu}^A$ and $\boldsymbol{\mu}^B$.

**Problem (A)** For $\boldsymbol{\mu}^A$, the optimal policy $\pi_A^*$ selects $S_2$ for every time. The corresponding cumulative reward is given by:

$$Rew_{\boldsymbol{\mu}^A}(\pi_A^*, T) = \sum_{n=1}^{\lceil \frac{2T}{3} \rceil} \frac{3n}{2T} + \frac{T}{3} . \tag{89}$$

For the given policy $\pi$, the cumulative reward is upper bounded as follows:

$$Rew_{\boldsymbol{\mu}^A}(\pi, T) = \frac{1}{2}\mathbb{E}_{\boldsymbol{\mu}_A, \pi}[N_{S_1, T}] + \sum_{n=1}^{T - \mathbb{E}_{\boldsymbol{\mu}_A, \pi}[N_{S_1, T}]} \mu_2^A(n) \tag{90}$$

$$\leq \frac{M}{2} + \sum_{n=1}^{T-M} \mu_2^A(n) \tag{91}$$

$$= \frac{M}{2} + \sum_{n=1}^{\lceil \frac{2T}{3} \rceil} \frac{3n}{2T} + \left(\frac{T}{3} - M\right) \tag{92}$$

$$= \sum_{n=1}^{\lceil \frac{2T}{3} \rceil} \frac{3n}{2T} + \frac{T}{3} - \frac{M}{2} , \tag{93}$$

where equation 91 holds since the cumulative reward is maximized as $\mathbb{E}_{\boldsymbol{\mu}^A, \pi}[N_{S_1, T}]$ minimized and it is guaranteed that $\mathbb{E}_{\boldsymbol{\mu}^A, \pi}[N_{S_1, T}] \geq \mathbb{E}_{\boldsymbol{\mu}^A, \pi}[N_{S_1, \frac{T}{3}}] = M$.

The cumulative regret is lower bounded by:

$$Reg_{\boldsymbol{\mu}^A}(\pi, T) \geq \sum_{n=1}^{\lceil \frac{2T}{3} \rceil} \frac{3n}{2T} + \frac{T}{3} - \left(\sum_{n=1}^{\lceil \frac{2T}{3} \rceil} \frac{3n}{2T} + \frac{T}{3} - \frac{M}{2}\right) \tag{94}$$

$$= \frac{M}{2} . \tag{95}$$

**Problem (B)** For $\boldsymbol{\mu}^B$, the optimal policy $\pi_B^*$ is selecting $S_1$, for every time. The corresponding cumulative reward is given by:

$$Rew_{\boldsymbol{\mu}^B}(\pi_B^*, T) = \frac{T}{2} . \tag{96}$$

For the given policy $\pi$, the cumulative reward is upper bounded as follows:

$$Rew_{\boldsymbol{\mu}^B}(\pi, T) = \frac{1}{2}\mathbb{E}_{\boldsymbol{\mu}_B, \pi}[N_{S_1, T}] + \sum_{n=1}^{T - \mathbb{E}_{\boldsymbol{\mu}_B, \pi}[N_{S_1, T}]} \mu_2^B(n) \tag{97}$$

$$\leq \frac{\left(\frac{2T}{3} + M\right)}{2} + \sum_{n=1}^{\lceil \frac{T}{3} - M \rceil} \mu_2^B(n) \tag{98}$$

$$= \sum_{n=1}^{\lceil \frac{T}{3} - M \rceil} \frac{3n}{2T} + \frac{T}{3} + \frac{M}{2} \tag{99}$$

$$= \frac{3}{4T}\left(\frac{T}{3} - M\right)\left(\frac{T}{3} - M + 1\right) + \frac{T}{3} + \frac{M}{2} \tag{100}$$

$$= \frac{3M^2}{4T} - \frac{3M}{4T} + \frac{5T}{12} + \frac{1}{4} , \tag{101}$$

where equation 91 holds since the cumulative reward is maximized as $\mathbb{E}_{\boldsymbol{\mu}_B, \pi}[N_{S_1, T}]$ maximized and it is guaranteed that $\mathbb{E}_{\boldsymbol{\mu}_B, \pi}[N_{S_1, T}] \leq \frac{2T}{3} + \mathbb{E}_{\boldsymbol{\mu}_B, \pi}[N_{S_1, \frac{T}{3}}] = \frac{2T}{3} + M$.

The cumulative regret is lower bounded by:

$$Reg_{\boldsymbol{\mu}^B}(\pi, T) \geq \frac{T}{2} - \left( \frac{3M^2}{4T} - \frac{3M}{4T} + \frac{5T}{12} + \frac{1}{4} \right) \tag{102}$$

$$= -\frac{3M^2}{4T} + \frac{3M}{4T} + \frac{T}{12} - \frac{1}{4} \tag{103}$$

$$\geq -\frac{3M^2}{4T} + \frac{3M}{4T} + \frac{T}{16} , \tag{104}$$

where equation 104 holds since we assume sufficiently large $T$.

From previous results, the worst-case regret can be lower bounded as follows:

$$\inf_{\pi} \sup_{\boldsymbol{\mu}} Reg_{\boldsymbol{\mu}}(\pi, T) \geq \inf_{\pi} \max \left\{ Reg_{\boldsymbol{\mu}^A}(\pi, T), Reg_{\boldsymbol{\mu}^B}(\pi, T) \right\} \tag{105}$$

$$= \inf_{M \in [0, \frac{T}{3}]} \max \left\{ \frac{M}{2}, -\frac{3M^2}{4T} + \frac{3M}{4T} + \frac{T}{16} \right\} \tag{106}$$

$$\geq \inf_{M \in [0, \frac{T}{3}]} \frac{-12M^2 + (8T + 12)M + T^2}{16T} \tag{107}$$

$$\geq \frac{T}{16} , \tag{108}$$

where equation 107 holds since $\max(a, b) \geq \frac{a+b}{2}$ and equation 108 holds since it is easily verified that equation 107 is minimized when $M = 0$, which completes the proof. □

Now, we expand Lemma 3 to general combinatorial setting. Let $L$ be an arbitrary constant. We define two problem $\boldsymbol{\mu}^{A,L}$ and $\boldsymbol{\mu}^{B,L}$ construct super arm set $\mathcal{S}_L$ as follows:

$$\mu_i^{A,L}(n) = \mu_i^{B,L}(n) = \frac{1}{2} \quad i \in [1, L] \tag{109}$$

$$\mu_i^{A,L}(n) = \begin{cases} \frac{3n}{2T} & \text{if } n \leq \frac{2T}{3} \\ 1 & \text{otherwise} \end{cases} \quad i \in [L+1, 2L] \tag{110}$$

$$\mu_i^{B,L}(n) = \begin{cases} \frac{3n}{2T} & \text{if } n \leq \frac{T}{3} \\ \frac{1}{2} & \text{otherwise} \end{cases} \quad i \in [L+1, 2L] \tag{111}$$

$$S_L := \{ (a_1, a_2, \ldots, a_L) : a_i \in \{i, L+i\} \quad i \in [L] \} . \tag{112}$$

Since it can be interpreted as solving $L$ independent problems, we have:

$$\inf_{\pi} \sup_{\boldsymbol{\mu}} Reg_{\boldsymbol{\mu}}(\pi, T) \geq \frac{LT}{16} . \tag{113}$$

□

## C.5 Proof of Theorem 5

*Proof.* We apply similar logic given in the Appendix C.4 to show that for the worst-case lower bound is $\Omega \left( \max \left\{ L\sqrt{T}, LT^{2-c} \right\} \right)$. Firstly, we consider non-combinatorial case.

**Lemma 4.** *Let $\mathcal{A}_c'$ be the subset of two-armed rising bandit problem with constraints given in equation 14 . For sufficiently large time $T$, any policy $\pi$ suffers regret:*

$$\min_{\pi} \max_{\boldsymbol{\mu} \in \mathcal{A}_c'} Reg_{\boldsymbol{\mu}}(\pi, T) \geq LT^{2-c} , \tag{114}$$

*Proof.* For convention, we define $\mu(m)$ and $F(m)$ as follows:

$$\mu(m) := \sum_{n=1}^{m} (n+1)^{-c} \tag{115}$$

$$F(m) := \sum_{n=1}^{m} \mu(n) . \tag{116}$$

Let $\boldsymbol{\mu}^A$ and $\boldsymbol{\mu}^B$ be two rising bandit instances. which are defined as:

$$\mu_1^A(n) = \mu_2^B(n) = \mu(P) - \varepsilon \tag{117}$$

$$\mu_2^A(n) = \mu(n) ; \tag{118}$$

$$\mu_2^B(n) = \begin{cases} \mu(n) & \text{if } n \leq P \\ \mu(P) & \text{otherwise} \end{cases} , \tag{119}$$

where $P = (2-c)^{\frac{1}{c-1}} T$ and $0 < \varepsilon < \mu(P)$ will be specified later. In this setting, we define $\mathcal{S}$ as follows:

$$\mathcal{S} = \{S_1, S_2\} , \tag{120}$$

where $S_1 = \{1\}$ and $S_2 = \{2\}$. Similar to Theorem 4, we define:

$$M := \mathbb{E}_{\boldsymbol{\mu}^A, \pi}[N_{S_1, P}] = \mathbb{E}_{\boldsymbol{\mu}^B, \pi}[N_{S_1, P}] \tag{121}$$

We note that $\boldsymbol{\mu}^A$ and $\boldsymbol{\mu}^B$ belongs to $\mathcal{A}_c'$. We firstly assume that the optimal super arm for $\boldsymbol{\mu}^A$ is $S_2$ and the optimal super arm for $\boldsymbol{\mu}^B$ is $S_1$. We will show that it is true after $\varepsilon$ is specified.

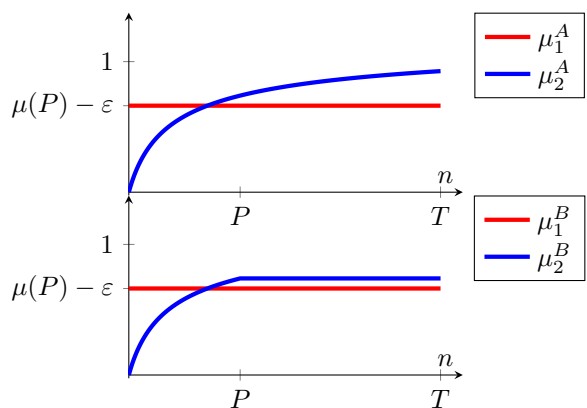

The main idea of the proof is that for any arbitrary policy $\pi'$, the agent receives the same rewards for both $\boldsymbol{\mu}^A$ and $\boldsymbol{\mu}^B$ at least until $P$, indicating that:

$$Rew_{\boldsymbol{\mu}^A}(\pi', P) = Rew_{\boldsymbol{\mu}^B}(\pi', P) . \tag{122}$$

**Problem (A)** For $\boldsymbol{\mu}^A$, the optimal policy $\pi_A^*$ is selecting $S_2$, for every time. The corresponding cumulative reward is given by:

$$Rew_{\boldsymbol{\mu}^A}(\pi_A^*, T) = F(T) . \tag{123}$$

For the given policy $\pi$, the cumulative reward is upper bounded as follows:

$$Rew_{\boldsymbol{\mu}^A}(\pi, T) = (\mu(P) - \varepsilon) \mathbb{E}_{\boldsymbol{\mu}^A, \pi}[N_{S_1, T}] + \sum_{n=1}^{T - \mathbb{E}_{\boldsymbol{\mu}^A, \pi}[N_{S_1, T}]} \mu_2^A(n) \tag{124}$$

$$\leq (\mu(P) - \varepsilon) M + F(T - M), \tag{125}$$

where equation 125 holds since the cumulative reward is maximized as $\mathbb{E}_{\boldsymbol{\mu}^A, \pi}[N_{S_1, T}]$ minimized and it is guaranteed that $\mathbb{E}_{\boldsymbol{\mu}^A, \pi}[N_{S_1, T}] \geq \mathbb{E}_{\boldsymbol{\mu}^A, \pi}[N_{S_1, P}]$.

With The cumulative regret is lower bounded by:

$$Reg_{\boldsymbol{\mu}^A}(\pi, T) \geq F(T) - (\mu(P) - \varepsilon) M - F(T - M) . \tag{126}$$

**Problem (B)** For $\boldsymbol{\mu}^B$, the optimal policy $\pi_B^*$ is selecting $S_1$, for every time. The corresponding cumulative reward is given by:

$$Rew_{\boldsymbol{\mu}^B}(\pi_B^*, T) = T(\mu(P) - \varepsilon) . \tag{127}$$

For the given policy $\pi$, the cumulative reward is upper bounded as follows:

$$Rew_{\boldsymbol{\mu}^B}(\pi, T) = (\mu(P) - \varepsilon)\, \mathbb{E}_{\boldsymbol{\mu}^B, \pi}[N_{S_1, T}] + \sum_{n=1}^{T - \mathbb{E}_{\boldsymbol{\mu}^B, \pi}[N_{S_1, T}]} \mu_2^B(n) \tag{128}$$

$$\leq (\mu(P) - \varepsilon)(T - P + M) + \sum_{n=1}^{P-M} \mu_2^B(n) \tag{129}$$

$$= (\mu(P) - \varepsilon)(T - P + M) + F(P - M) , \tag{130}$$

where equation 129 holds since the cumulative reward is maximized as $\mathbb{E}_{\boldsymbol{\mu}^B, \pi}[N_{S_1, T}]$ maximized and it is guaranteed that $\mathbb{E}_{\boldsymbol{\mu}^B, \pi}[N_{S_1, T}] \leq T - P + \mathbb{E}_{\boldsymbol{\mu}^B, \pi}[N_{S_1, P}] = T - P + M$.

The cumulative regret is lower bounded by:

$$Reg_{\boldsymbol{\mu}^B}(\pi, T) \geq (\mu(P) - \varepsilon)T - (\mu(P) - \varepsilon)(T - P + M) - F(P - M) \tag{131}$$

$$= (\mu(P) - \varepsilon)(P - M) - F(P - M) . \tag{132}$$

From previous results, the worst-case regret can be lower bounded as follows:

$$\inf_{\pi} \sup_{\boldsymbol{\mu}} Reg_{\boldsymbol{\mu}}(\pi, T) \tag{133}$$

$$\geq \inf_{\pi} \max \left\{ Reg_{\boldsymbol{\mu}^A}(\pi, T), Reg_{\boldsymbol{\mu}^B}(\pi, T) \right\} \tag{134}$$

$$= \inf_{M \in [0, P]} \max \left\{ F(T) - (\mu(P) - \varepsilon)M - F(T - M), (\mu(P) - \varepsilon)(P - M) - F(P - M) \right\} \tag{135}$$

$$\geq \inf_{M \in [0, P]} \frac{F(T) - F(T - M) - F(P - M) + (\mu(P) - \varepsilon)(P - 2M)}{2} , \tag{136}$$

where equation 136 holds since $\max(a, b) \geq \frac{a+b}{2}$. We observe that equation 136 is unimodal over $P$, which means that it increases to a maximum value and then decreases. More precisely, let $A(n) := F(T) - F(T - n) - F(P - n) + (\mu(P) - \varepsilon)(P - 2n)$. Then, we have:

$$A(n+1) - A(n) = F(T - n) - F(T - n + 1) + F(P - n) - F(P - n + 1) - 2(\mu(P) - \varepsilon) \tag{137}$$

$$= \mu(T - n + 1) + \mu(P - n + 1) - 2(\mu(P) - \varepsilon), \tag{138}$$

which means that $A(n)$ is concave, which means that $A(n)$ is unimodal. It implies that:

$$\inf_{M \in [0, P]} \left( \frac{F(T) - F(T - M) - F(P - M) + (\mu(P) - \varepsilon)(P - 2M)}{2} \right) \tag{139}$$

$$\geq \min \left\{ \frac{(\mu(P) - \varepsilon)P - F(P)}{2}, \frac{F(T) - F(T - P) - (\mu(P) - \varepsilon)P}{2} \right\} . \tag{140}$$

equation 140 consists of two terms: the first term is obtained by setting $M = 0$ and the second term is obtained by setting $M = P$.

To calculate two terms, we use the property of monotone functions.

**Proposition 1.** *If $a$ and $b$ are integers with $a < b$ and $f$ is some real-valued function monotone on $[a, b]$, we have:*

$$\min\{f(a), f(b)\} \leq \sum_{n=a}^{b} f(n) - \int_a^b f(t)\, dt \leq \max\{f(a), f(b)\} . \tag{141}$$

Proposition 1 indicates that we can bound $\mu(n)$ and $F(n)$ as follows:

$$\mu(n) \leq \int_{x=1}^n (x+1)^{-c} dx + 2^{-c} \tag{142}$$

$$= \frac{1}{c-1} \left( 2^{1-c} - (n+1)^{1-c} \right) + 2^{-c} . \tag{143}$$

For simplicity, we denote $(2-c)^{\frac{1}{c-1}}$ by $a$ so that $P = aT$. Then, we have:

$$(\mu(P) - \varepsilon)P - F(P) \tag{144}$$

$$= P\sum_{n=1}^{P}(n+1)^{-c} - P\varepsilon - \sum_{n=1}^{P}(P+1-n)(n+1)^{-c} \tag{145}$$

$$= \sum_{n=1}^{P}(n-1)(n+1)^{-c} - P\varepsilon \tag{146}$$

$$\geq P^{2-c} - P\varepsilon \tag{147}$$

$$= (aT)^{2-c} - (aT)\varepsilon . \tag{148}$$

Similarly, we have:

$$F(T) - F(T-P) - (\mu(P) - \varepsilon)P \tag{149}$$

$$= \sum_{n=1}^{T}(T+1-n)(n+1)^{-c} - \sum_{n=1}^{T-P}(T-P+1-n)(n+1)^{-c} - (\mu(P)-\varepsilon)P \tag{150}$$

$$= \sum_{n=T-P+1}^{T}(T+1-n)(n+1)^{-c} + \sum_{n=1}^{T-P}P(n+1)^{-c} - (\mu(P)-\varepsilon)P \tag{151}$$

$$\geq P(T+1)^{-c} + (T-P)P(T+P-1)^{-c} - (\mu(P)-\varepsilon)P \tag{152}$$

$$= aT(T+1)^{-c} + (T-aT)aT(T+aT-1)^{-c} - \left(\frac{2^{1-c}}{c-1} - \frac{(aT+1)^{1-c}}{c-1} + 2^{-c} - \varepsilon\right)aT \tag{153}$$

$$= c_2 T^{2-c} + o(T^{2-c}) + \varepsilon aT , \tag{154}$$

where Now, we define $\varepsilon$ so that equation 148 equals equation 154:

$$2aT\varepsilon = (c_2 + a^{2-c})T^{2-c} + o(T^{2-c}) \tag{155}$$

Then, by substituting $\varepsilon$ to equation 148 and equation 154, we have:

$$Reg_{\boldsymbol{\mu}}(\pi, T) \geq \Omega\left(T^{2-c}\right) . \tag{156}$$

$\square$

Now, we expand Lemma 4 to general combinatorial setting. Let $L$ be an arbitrary constant. As before, we define two problem $\boldsymbol{\mu^{A,L}}$ and $\boldsymbol{\mu^{B,L}}$ construct super arm set $\mathcal{S}_L$ as follows:

$$\mu_i^{A,L}(n) = \mu_i^{B,L}(n) = \mu(P) - \varepsilon, \quad i \in [L], \tag{157}$$

$$\mu_i^{A,L}(n) = \mu(n), \quad i \in [L+1, 2L], \tag{158}$$

$$\mu_i^{B,L}(n) = \begin{cases} \mu(n) & \text{if } n \leq P \\ \mu(P) & \text{otherwise} \end{cases} \quad i \in [L+1, 2L] , \tag{159}$$

$$S_L := \{(a_1, a_2, \ldots, a_L) : a_i \in \{i, L+i\} \quad i \in [L]\} . \tag{160}$$

Due to same reason in Appendix C.4 we have:

$$\inf_{\pi} \sup_{\boldsymbol{\mu} \in \mathcal{A}_c} Reg_{\boldsymbol{\mu}}(\pi, T) \geq \Omega(LT^{2-c}) . \tag{161}$$

Now, we note that any stationary bandit problem is included in $\mathcal{A}_c$, since $\gamma_i(n) = 0$ for all base arm $i \in [K]$. Previous literature has proven that for stationary bandit problem, the worst-case regret lower bound is $\Omega(\sqrt{KT})$ (Lattimore & Szepesvári, 2020). Similarly, we can extend this setting to combinatorial setting:

$$\inf_{\pi} \sup_{\boldsymbol{\mu} \in \mathcal{A}_c} Reg_{\boldsymbol{\mu}}(\pi, T) \geq \Omega(L\sqrt{T}) . \tag{162}$$

Combining these results, we conclude:

$$\min_{\pi} \max_{\boldsymbol{\mu} \in \mathcal{A}_c} Reg_{\boldsymbol{\mu}}(\pi, T) \geq \Omega\left(\max\left\{L\sqrt{T}, LT^{2-c}\right\}\right) . \tag{163}$$

$\square$

## D  PSEUDOCODE AND DESCRIPTION OF BASELINES

In Section 6, we have considered 5 baseline algorithms to evaluate CRUCB's performance. Each algorithm is carefully chosen to highlight different aspects of the bandit problem, such as rising rewards and combinatorial settings. In this section, we provide the pseudocode and detailed descriptions for each baseline algorithm.

### D.1  R-ED-UCB (METELLI ET AL., 2022)

R-ed-UCB is a rising bandit algorithm that employs a sliding-window approach combined with UCB-based optimistic reward estimation algorithm, specifically designed for rising rewards. While it shares the core estimation method ($\acute{\mu}_i(t)$) with CRUCB, R-ed-UCB applies this method directly to super arms and selects the maximum one, rather than applying it to base arms and solving the combinatorial problem as in CRUCB. R-ed-UCB would be less effective in complex environments where the number of super arms significantly exceeds the number of base arms, as it does not benefit from the shared exploration of common base arms, leading to reduced exploration efficiency.

---

**Algorithm 2** `Rested UCB (R-ed-UCB)`

---

**Input** $N_{i,0} \leftarrow 0$ for all $i \in [|\mathcal{S}|]$, Sliding window parameter $\varepsilon$.
**Initialize** Play each super arm $S_i$ two times for each $i \in [|\mathcal{S}|]$.
**for** $t \in (1, \ldots, T)$ **do**
    Calculate Future-UCB $\acute{\mu}_i(t)$ for each super arm.
    $S_t \leftarrow \texttt{Solver}(\acute{\mu}_1(t), \acute{\mu}_2(t), \cdots, \acute{\mu}_{|\mathcal{S}|}(t))$.
    Play $S_t$ and observe reward $R_t$.
    Update $\mathcal{F}_t$ and $N_{i,t}$.
**end for**

---

### D.2  SW-UCB (GARIVIER & MOULINES, 2011)

SW-UCB is a non-stationary bandit algorithm that uses a sliding-window approach with UCB algorithm. It estimates the reward of each super arm and confidence bounds using the following expressions:

$$\hat{\mu}_i^{\text{SW-UCB}}(t) := \frac{1}{h} \sum_{l=N_{i,t-1}-h+1}^{N_{i,t-1}} X_i(l) \tag{164}$$

$$\beta_i^{\text{SW-UCB}}(t) := \sqrt{\frac{3 \log t}{2 N_{i,t-1}}} \tag{165}$$

$$\acute{\mu}_i^{\text{SW-UCB}}(t) := \hat{\mu}_i^{\text{SW-UCB}}(t) + \beta_i^{\text{SW-UCB}}(t) . \tag{166}$$

While the SW-UCB algorithm is similar to R-ed-UCB, it differs slightly in the values it estimates. Additionally, SW-UCB uses a fixed sliding window size, in contrast to the dynamic sliding window size employed by R-ed-UCB. Similar to R-ed-UCB, SW-UCB would be less effective in complex environments.

---

**Algorithm 3** `Sliding Window-UCB (SW-UCB)`

---

**Input** $N_{i,0} \leftarrow 0$ for all $i \in [|\mathcal{S}|]$, Sliding window size $h$.
**Initialize** Play each super arm $S_i$ two times for each $i \in [|\mathcal{S}|]$.
**for** $t \in (1, \ldots, T)$ **do**
    For each super arm $S_i$, set $\acute{\mu}_i^{\text{SW-UCB}}(t) = \hat{\mu}_i^{\text{SW-UCB}}(t) + \beta_i^{\text{SW-UCB}}(t)$.
    $S_t \leftarrow \arg\max(\acute{\mu}_1^{\text{SW-UCB}}(t), \acute{\mu}_2^{\text{SW-UCB}}(t), \cdots, \acute{\mu}_{|\mathcal{S}|}^{\text{SW-UCB}}(t))$.
    Play $S_t$ and observe reward $X_{S_t}(t)$.
    Update $\hat{\mu}_i^{\text{SW-UCB}}(t)$ and $N_{i,t}$.
**end for**

---

### D.3    SW-CUCB (CHEN ET AL., 2021)

SW-CUCB is a non-stationary combinatorial bandit algorithm that uses a sliding-window approach with UCB algorithm for combinatorial setting. It estimates the values $\hat{\mu}_i^{\text{SW-CUCB}}(t)$ and $\beta_i^{\text{SW-CUCB}}(t)$, which are nearly identical to those used in SW-UCB but specifically adapted for base arms. SW-CUCB then utilizes Solver to address the combinatorial problem.

---

**Algorithm 4** `Sliding Window-Combinatorial UCB (SW-CUCB)`

---

**Input** $N_{i,0} \leftarrow 0$ for all $i \in [K]$, Sliding window size $h$.
**Initialize** Play arbitrary super arm including base arm $i$ two times for each $i \in [K]$.
**for** $t \in (1, \ldots, T)$ **do**
    For each base arm $i$, set $\acute{\mu}_i^{\text{SW-CUCB}}(t) = \hat{\mu}_i^{\text{SW-CUCB}}(t) + \beta_i^{\text{SW-CUCB}}(t)$.
    $S_t \leftarrow \text{Solver}(\acute{\mu}_1^{\text{SW-CUCB}}(t), \acute{\mu}_2^{\text{SW-CUCB}}(t), \cdots, \acute{\mu}_K^{\text{SW-CUCB}}(t))$.
    Play $S_t$ and observe reward $X_{S_t}(t)$.
    Update $\hat{\mu}_i^{\text{SW-CUCB}}(t)$ and $N_{i,t}$.
**end for**

---

### D.4    SW-TS (TROVO ET AL., 2020)

SW-TS is a non-stationary bandit algorithm that uses a sliding-window approach with Thompson Sampling. Since outcomes are bounded, the algorithm updates the parameters by adds $X_{S_t}(t)$ to $\alpha$ and $1 - X_{S_t}(t)$ to $\beta$ based on the observed output $X_{S_t}(t)$. SW-TS also utilizes a fixed sliding window size similar to SW-UCB. Similar to R-ed-UCB and SW-UCB, SW-TS also operates directly on super arms, it may suffer from reduced exploration efficiency in complex environments.

---

**Algorithm 5** `Sliding Window Thompson Sampling (SW-TS)`

---

**Input** Sliding window size $h$.
**Initialize** Play each super arm $S_i$ two times for each $i \in [|\mathcal{S}|]$.
**for** $t \in (1, \ldots, T)$ **do**
    For each super arm $S_i$, set $\theta_i(t) \sim Beta(\alpha_i + 1, \beta_i + 1)$.
    $S_t \leftarrow \arg\max(\theta_1(t), \theta_2(t), \cdots, \theta_K(t))$.
    Play $S_t$ and observe reward $X_{S_t}(t)$.
    Update $\alpha_i$ and $\beta_i$.
**end for**

---

### D.5    SW-CTS

SW-CTS is a non-stationary combinatorial bandit algorithm that uses a sliding-window approach with Thompson Sampling for combinatorial setting. While it operates similarly to SW-TS, the key difference is that SW-CTS performs estimation at base arms then solves the combinatorial problem using Solver.

---

**Algorithm 6** `Sliding Window-Combinatorial Thompson Sampling (SW-CTS)`

---

**Input** Sliding window size $h$.
**Initialize** Play arbitrary super arm including base arm $i$ two times for each $i \in [K]$.
**for** $t \in (1, \ldots, T)$ **do**
    For each base arm $i$, set $\theta_i(t) \sim Beta(\alpha_i + 1, \beta_i + 1)$.
    $S_t \leftarrow \text{Solver}(\theta_1(t), \theta_2(t), \cdots, \theta_K(t))$.
    Play $S_t$ and observe reward $X_{S_t}(t)$.
    Update $\alpha_i$ and $\beta_i$.
**end for**

---

# E    EXPERIMENTAL DETAILS

Table 1: **Overview of task specifications.** We summarizes the details of each task, including the number of base arms $K$, the number of super arms $|\mathcal{S}|$, and the maximal size of a super arm $L$.

| Environment | Experiment | difficulty | $K$ | $|\mathcal{S}|$ | $L$ |
|---|---|---|---|---|---|
| Synthetic environments (Section 6.1 & Appendix F) | Online shortest path | toy | 3 | 2 | 2 |
| | | easy | 12 | 6 | 4 |
| | | complex | 60 | 252 | 10 |
| | Maximum weighted matching | easy | 8 | 12 | 2 |
| | | complex | 28 | 840 | 4 |
| | Minimum spanning tree | easy | 6 | 8 | 3 |
| | | complex | 15 | 1296 | 5 |
| | $k$-MAX | - | 5 | 10 | 2 |
| Deep reinforcement learning (Section 6.2) | AntMaze-easy | - | 7 | 3 | 5 |
| | AntMaze-complex | - | 48 | 178 | 15 |

Table 2: **Hyperparameters for AntMaze Tasks.**

| | AntMaze-easy | AntMaze-complex |
|---|---|---|
| number of graph nodes | 6 | 16 |
| fail condition | 100 | 100 |
| maximum length of episode | 500 | 1000 |
| $T$ | 2000 | 3000 |
| hidden layer | (256, 256) | (256, 256) |
| actor lr | 0.0001 | 0.0001 |
| critic lr | 0.001 | 0.001 |
| $\tau$ | 0.005 | 0.005 |
| $\gamma$ | 0.99 | 0.99 |
| batch size | 1024 | 1024 |

In Section 6, we conduct experiments in two distinct environments: synthetic environments and deep reinforcement learning settings. This section provides a detailed description of each environment, including their design and hyperparameters. The specifications for each experiment are summarized in Table 1.

## E.1    SYNTHETIC ENVIRONMENTS

In the synthetic environments, we have the flexibility to design reward functions by choosing arbitrary values. Here, we set $c = 1.2$, which lies in the range between 1 and 1.5. This choice is motivated by the theoretical reasoning discussed in Section 5. To be specific, $\gamma(n) = \left(\left[\frac{n}{1000} + 1\right] \cdot 1000 + 1\right)^{-1.2}$ for $n < 20000$ and 0 for $n \geq 20000$ with $\sigma = 0.01$. In the simpler environments, we use $\gamma(n) = \left(\left[\frac{n}{250} + 1\right] \cdot 250 + 1\right)^{-1.2}$ for $n < 5000$ and 0 for $n \geq 5000$. As depicted in Figure 3, the regret upper bound for these environments is $O(T^{\frac{1}{1.2}})$, and the regret lower bound is $O(T^{0.8})$. Therefore, while the regret observed in Figure 5b appears nearly linear, which aligns with the theoretical bounds, it still demonstrates superior performance compared to other baseline algorithms.

## E.2    DEEP REINFORCEMENT LEARNING

In the deep reinforcement learning environments, we conducted experiments using the AntMaze environment. AntMaze is a hierarchical goal conditioned reinforcement learning task where an ant robot navigates to a predefined goal hierarchically. The ant robot in this environment has four legs,

each with two joints, resulting in an action space that controls a total of eight joints. The reward structure for the low-level agent is simple: the agent receives a reward of 0 when it reaches the goal or comes within a certain distance of it, and a reward of -1 otherwise. Our experiments are carried out in the scenario depicted in Figure 5, which shows the map and corresponding graph structure used. To ensure consistent and repeated exploration over the fixed graph, we utilized a code based on the algorithm described in (Yoon et al., 2024) without the adaptive grid refinement. The hyperparameters used in these experiments are summarized in Table 2.

In each experiment, the algorithm generates a path that the ant robot follows, receiving feedback based on success or failure. For combinatorial methods, the agent does not persist with a single edge until the episode ends; if the agent fails to reach the goal within 100 steps, the attempt is considered a failure. In this case, the reward is set to 0, and the agent fails to attempt the next edge, which known as the cascading bandit setting. If the agent successfully reaches the goal, the reward is proportional to the efficiency, calculated as the number of steps taken divided by 100. For non-combinatorial methods, the reward for success is determined by the number of steps taken divided by the maximum length of the episode. We note that while the reward function of AntMaze is non-concave, as depicted in Figure 7a, and cascading bandit setting, we confirmed that RCUCB performs well, as illustrated in Figure 7.

# F   ADDITIONAL EXPERIMENTS

In this section, we present additional experiments to evaluate the performance of CRUCB in a broader set of environments. Specifically, we test CRUCB on three representative combinatorial optimization problems, $k$-MAX (Section F.1), maximum weighted matching (Section F.2), and minimum spanning tree (Section F.3). These experiments demonstrate that CRUCB maintains strong performance across diverse scenarios, further validating its robustness and adaptability.

## F.1   $k$-MAX TASK

We investigate the $k$-MAX setting, where the reward is determined by the maximum value among outcomes of the selected base arm. As shown in Theorem 1, the optimal policy for the $k$-MAX may not always involve consistently pulling a single super arm. However, since the $k$-MAX satisfies the additive-bounded reward assumption in Theorem 2 with $B_L = \frac{1}{\sqrt{k}}, B_U = 1$, we use an approximate optimal constant policy (consistntly pulling $(1, 5)$) to calculate regret.

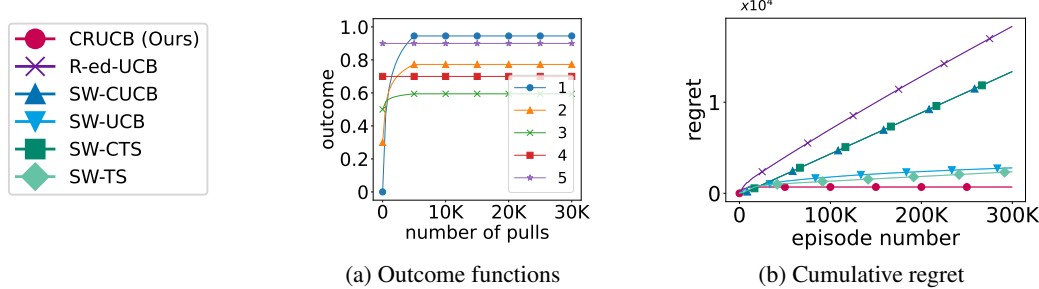

(a) Outcome functions     (b) Cumulative regret

Figure 9: $k$-**MAX task.** (a) Reward functions ($c = 1.2$) for base arms 1–5, where $K = 5$ and $k = 2$. (b) Regret curves for $K$-MAX. Lines show average; shaded areas indicate 99% confidence intervals over 5 runs.

The results, as shown in Figure 9b, demonstrate that CRUCB consistently outperforms other algorithms. R-ed-UCB shows sub-optimal regret due to the *partially shared enhancement*. Notably, we observe that among non-stationary algorithms, combinatorial algorithms (SW-CUCB, SW-CTS) perform worse than non-combinatorial algorithms (SW-UCB, SW-TS). Non-combinatorial algorithms select the early peaker $(5)$ frequently while evenly exploring other edges. On the other hand, combinatorial algorithms select early peakers $(4, 5)$, limiting exploration of late bloomers and preventing them from fully rising their potential.

## F.2 MAXIMUM WEIGHTED MATCHING

We conduct experiments on maximum weighted matching task, a widely studied classic combinatorial optimization problem. In this task, we are given two disjoint sets of nodes, $U$ and $V$, and the goal is to find a matching where each node $u_i \in U$ is paired with a unique node $v_j \in V$, ensuring no overlapping connections. The objective is to maximize the total reward by selecting the best set of edges between these nodes.

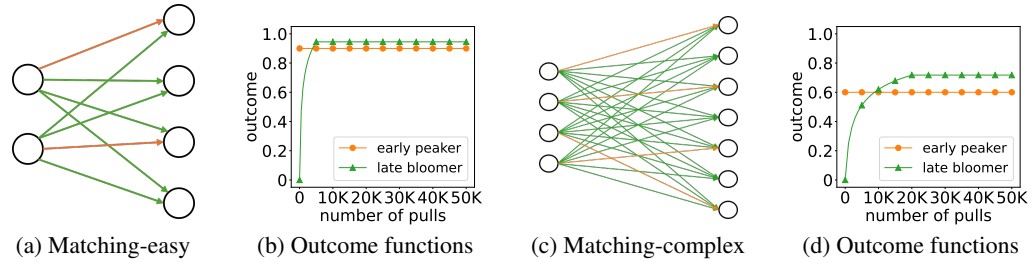

(a) Matching-easy    (b) Outcome functions    (c) Matching-complex    (d) Outcome functions

Figure 10: **Maximum weighted matching task.** (a, c) Graphs used to evaluate CRUCB and baselines. (b, d) Corresponding outcome functions for each task.

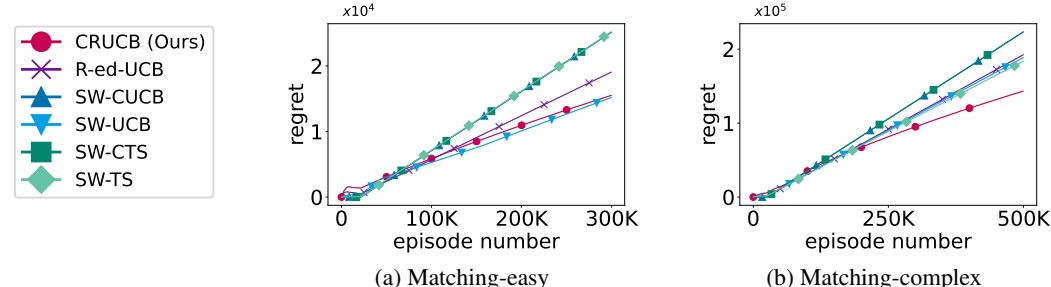

(a) Matching-easy    (b) Matching-complex

Figure 11: **Cumulative regret in maximum weighted matching task.** Regret curves for (a) Matching-easy and (b) Matching-complex. Lines show average; shaded areas indicate 99% confidence intervals over 5 runs.

We use the same outcome function as in the online shortest path problem, shown in Figure 10b and d. The graph structures are depicted in Figure 10a and c. The regret results, shown in Figure 11, confirm that CRUCB outperforms the baseline algorithms in this task.

The task is particularly relevant in settings like job matching, where each job can be matched to a worker, and the reward might increase over time as workers gain experience. This makes the problem a perfect fit for combinatorial bandit settings, where the rewards of certain matches (such as experienced workers with higher skill levels) rise as more interactions occur, highlighting the rising aspect of the task.

### F.3 MINIMUM SPANNING TREE

We conduct experiments on minimum spanning tree task, a fundamental problem in combinatorial optimization, where the objective is to find a subset of edges that connect all nodes in a graph with the minimum total edge weight, ensuring no cycles. However, in our setting, we treat the weight of each edge as a $1-$outcome, meaning we aim to maximize the total outcome, which is equivalent to minimizing the total edge weight.

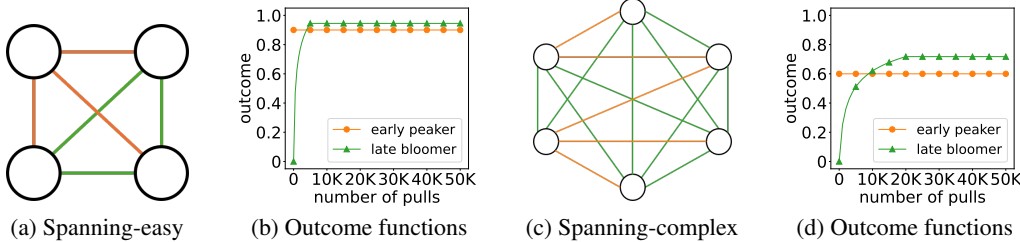

(a) Spanning-easy  (b) Outcome functions  (c) Spanning-complex  (d) Outcome functions

Figure 12: **Minimum spanning tree task.** (a, c) Graphs used to evaluate CRUCB and baselines. (b, d) Corresponding outcome functions for each task.

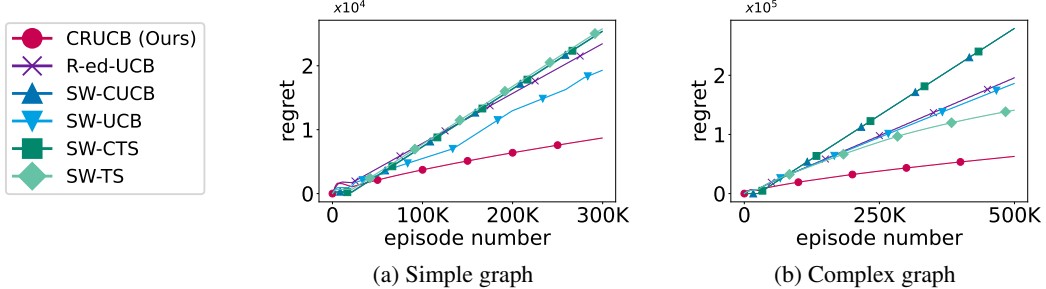

(a) Simple graph  (b) Complex graph

Figure 13: **Cumulative regret in minimum spanning tree task.** Regret curves for (a) Spanning-easy and (b) Spanning-complex. Lines show average; shaded areas indicate 99% confidence intervals over 5 runs.

Similarly, we evaluate minimum spanning tree task with the same outcome function from Figure 12b and d, applied to the graph structures in Figure 12a and c. The regret results, presented in Figure 13, indicate that CRUCB consistently performs better than the baselines.

This formulation is particularly relevant in practical applications such as network routing, where the objective is to establish efficient communication across a distributed system. Over time, as certain paths are used more frequently, the network can adapt and optimize its behavior: caches warm up, congestion reduces through load balancing, and routing protocols fine-tune their decisions. As a result, the effective cost of using the same edge decreases, which translates into a rising reward for that edge.

### F.4 SENSITIVITY TO THE WINDOW SIZE PARAMETER

To evaluate the robustness of CRUCB to the choice of the window size hyperparameter $\epsilon$, we conducted a sensitivity analysis in the Path-easy environment (described in Section 6.1). We compared the cumulative regret at different episode counts for $\epsilon$ values of 0.05, 0.125, 0.25, and 0.4.

Table 3: Cumulative regret at different episodes for various $\epsilon$ values in the Path-easy task.

| **Regret** | $\epsilon = 0.05$ | $\epsilon = 0.125$ | $\epsilon = 0.25$ | $\epsilon = 0.4$ |
|---|---|---|---|---|
| 100K | 8019.34 | 8020.53 | 8020.53 | 8019.64 |
| 200K | 13715.91 | 13717.10 | 13717.10 | 13716.21 |
| 300K | 19060.71 | 19061.90 | 19061.90 | 19061.01 |

The results, summarized in the table below, show that the performance of CRUCB is remarkably stable across this wide range of values, indicating that our algorithm is not overly sensitive to this hyperparameter choice in practice.

# G   HEATMAPS IN ANTMAZE-COMPLEX

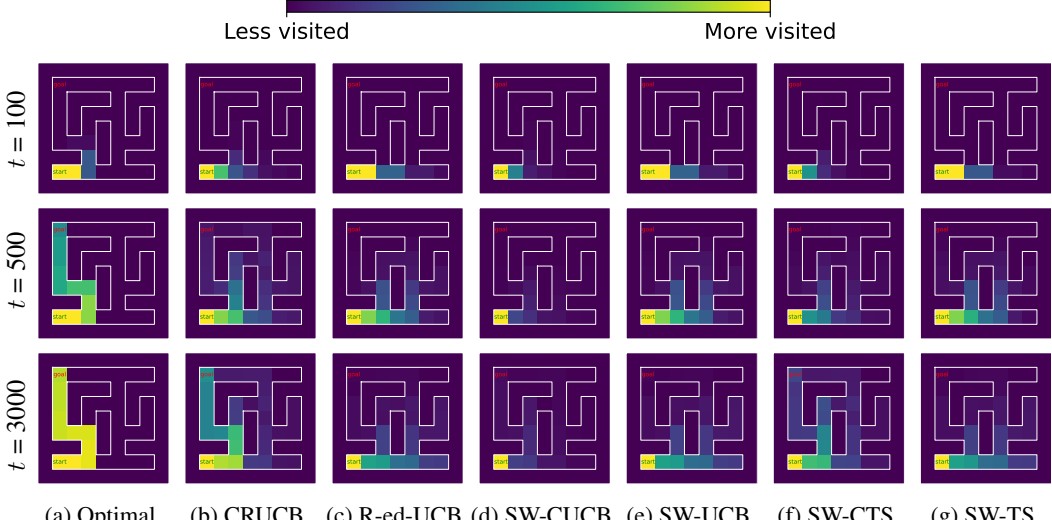

Figure 14: **Heatmap illustrating visit frequencies in AntMaze-complex.** We visualize the visit frequencies for the optimal policy, CRUCB, and baseline algorithms. The heatmap includes three rows representing visit frequencies until episode numbers 100, 500, and 3000.

In Figure 14, we provide a more comprehensive view by illustrating the exploration patterns across all baseline algorithms at various stages. The optimal policy, which follows the oracle constant policy, only explores the optimal path from the start to the goal, resulting in highly focused exploration along this path, as depicted in Figure 14a. In Figure 14b, CRUCB exhibits exploration patterns most similar to the optimal policy compared to other baselines, demonstrating its efficiency in targeting the goal effectively. Among the baselines, SW-CTS notably aligns most closely with the optimal policy in the exploration patterns and is the only algorithm to show a significant difference in regret compared to the others, as seen in Figure 7c. In comparison, algorithms not specifically designed for combinatorial settings, such as R-ed-UCB, SW-UCB, and SW-TS, suffer from less efficient exploration. Their exploration resembles a breadth-first search pattern, as they must explore a broader range of super arms despite having a given goal.

In Figure 15 and Figure 16, we further analyze the exploration behaviors of each algorithm by visualizing their try frequencies at both the path and edge levels. The optimal policy concentrates its tries exclusively along the shortest path, resulting in highly localized activity in both visualizations. CRUCB exhibits exploration patterns that closely resemble those of the optimal policy, maintaining focused and structured exploration throughout. Notably, AntMaze-complex includes 178 possible paths, which makes exhaustive exploration highly time-consuming. As illustrated in Figure 15, non-combinatorial algorithms struggle with this complexity: by episode 100, some paths remain untried, and even by episode 3000, their exploration remains broadly distributed and unguided, indicating inefficient use of the exploration budget.

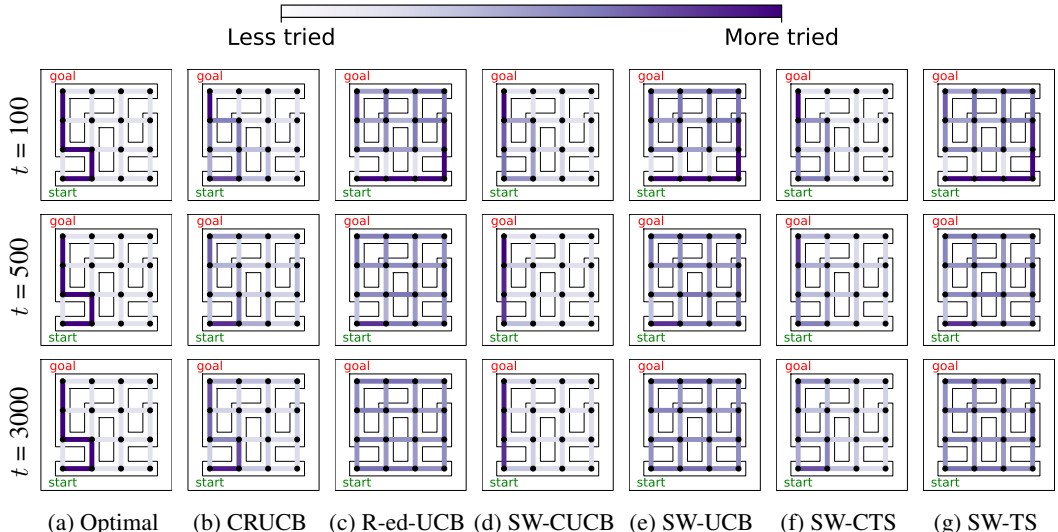

Figure 15: **Heatmap illustrating the path-level try frequencies in AntMaze-complex.** We visualize the path-level try frequencies for the optimal policy, CRUCB, and baseline algorithms. The heatmap includes three rows representing the path-level try frequencies until episode numbers 100, 500, and 3000.

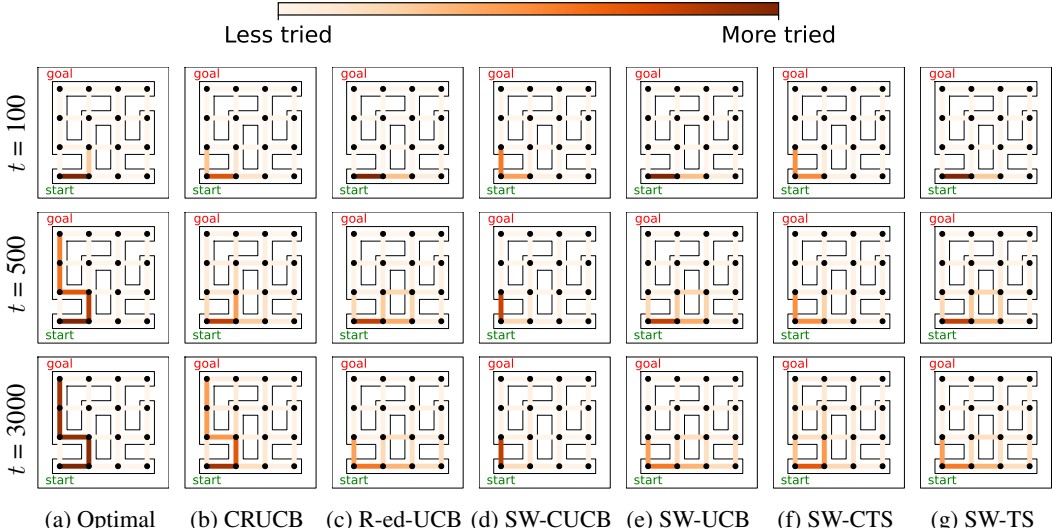

Figure 16: **Heatmap illustrating the edge-level try frequencies in AntMaze-complex.** We visualize the edge-level try frequencies for the optimal policy, CRUCB, and baseline algorithms. The heatmap includes three rows representing the edge-level try frequencies until episode numbers 100, 500, and 3000.

## H    THE USE OF LARGE LANGUAGE MODELS (LLMS)

In the preparation of this manuscript, we utilized a large language model (LLM) as a writing assistant to aid in polishing the text. The LLM's role was limited to refining phrasing and grammar in author-written drafts, suggesting alternative sentence structures to improve clarity, and helping maintain a consistent academic tone. All technical contributions, theoretical results, experimental designs, and final claims were conceived and developed solely by the human authors. The authors thoroughly reviewed and edited the manuscript and take full responsibility for all content presented in this paper.

