# OpenReview forum: "Combinatorial Rising Bandits"
_ICLR.cc/2026/Conference — ICLR 2026 Poster_

### Official Review · Reviewer_zJJk · 2025-10-25

**Soundness:** 2
**Presentation:** 3
**Contribution:** 2
**Rating:** 4
**Confidence:** 2

**Summary:**

This paper studies combinatorial semi-bandits with rested rising rewards, a setting relevant to diverse application domains such as robotics and maintenance scheduling. To address this problem, the authors propose the CRUCB algorithm and provide corresponding regret guarantees. They further demonstrate the effectiveness of the method through empirical evaluations.

**Strengths:**

This paper is the first to consider combinatorial semi-bandits with rising rewards.

**Weaknesses:**

1. My main concern is the novelty of the proposed algorithm. The paper does not clearly identify what non-trivial challenges arise specifically from incorporating rising rewards into the combinatorial semi-bandit setting, nor how the approach fundamentally differs from existing methods for rising or combinatorial bandits.

2. It is unclear whether the presented regret bound is tight.

3. The role of window sizes and their effect on performance is not discussed. Providing guidance or theoretical justification for the choice of window size would strengthen the practical relevance of the method.

**Questions:**

1. Could the authors elaborate on the unique challenges of incorporating rising rewards into the combinatorial semi-bandit setting? It is not fully clear what makes this extension technically difficult compared to standard combinatorial or rising bandits.

2. The novelty of the proposed algorithm relative to prior work on rising bandits is not fully articulated. A more explicit comparison against existing methods (e.g., in the algorithm section) would better highlight the contribution.

3. The discussion on effective window size remains insufficient. In particular, the choices of window parameters for equations (10), (11), and (12) are not explained, and it is unclear how these influence theoretical guarantees or empirical performance.

4. Regarding the experimental setup:
   - How were the window sizes selected for the sliding-window type baselines?
   - How was  R-ed-UCB (with bandit feedback) adapted to the combinatorial semi-bandit feedback model?

---

> ### Author Response · Authors · 2025-11-17
>
> Dear Reviewer zJJk,
>
> We are very grateful for your thorough review and for your specific, constructive comments. We appreciate the time you dedicated to our work, and your feedback on technical novelty, regret tightness, and hyperparameter choices has been invaluable in helping us identify areas that needed clearer articulation. We have carefully considered all your points and provide our responses below.
>
> ---
>
> > **Weakness 1 & Question 1& Question 2.**  My main concern is the novelty of the proposed algorithm. The paper does not clearly identify what non-trivial challenges arise specifically from incorporating rising rewards into the combinatorial semi-bandit setting, nor how the approach fundamentally differs from existing methods for rising or combinatorial bandits.
> Could the authors elaborate on the unique challenges of incorporating rising rewards into the combinatorial semi-bandit setting? It is not fully clear what makes this extension technically difficult compared to standard combinatorial or rising bandits.
> The novelty of the proposed algorithm relative to prior work on rising bandits is not fully articulated. A more explicit comparison against existing methods (e.g., in the algorithm section) would better highlight the contribution.
>
> Thank you for the opportunity to clarify our work's technical challenges and novelty, which we view in two primary areas.
>
> First, our core novelty is the formalization and deep structural analysis of a new problem class (CRB) with a unique, emergent challenge we term **partially shared enhancement.** You asked what non-trivial challenges arise from this combination, and this is the key. It is not a simple extension, as the challenge fundamentally alters the problem's structure as we prove in Theorem 1. It shows that the optimal policy itself changes and becomes far more complex than in prior bandit settings. Critically, our further analysis in Theorem 2 then makes this problem tractable by characterizing when a simpler constant policy still serves as a good approximation. Therefore, our contribution is this deep analysis of a new and challenging problem, not a simple mix of existing methods. We believe Appendix B directly addresses this point by providing a detailed comparison that articulates how these unique challenges distinguish our framework from prior rising bandit literature.
>
> Second, a key aspect of our novelty is establishing a crucial bridge between bandit theory and its application in mainstream AI research. To move beyond synthetic environments, we are the first to demonstrate that our CRB framework naturally models the hierarchical decision-making process within the widely-used AntMaze benchmark. This validation in a complex, mainstream RL environment is a significant contribution. It confirms that CRB is not merely a theoretical construct but a relevant and applicable model for modern AI challenges, directly connecting our theoretical advances to the practical problems faced by the broader research community.
>
> ---
>
> > **Weakness 2.** It is unclear whether the presented regret bound is tight.
>
> Thank you for this crucial point. We rigorously analyzed the tightness of our bound and dedicated Section 5.2 to this question. In rising bandit settings, the optimal regret rate depends on the specific reward growth curve, making a single universal lower bound insufficient. To provide a meaningful analysis, we considered a canonical class of problems parameterized by difficulty $c$ (where growth is bounded by $(n+1)^{-c}$). For this class, we derived both an upper bound (Corollary 2) and a corresponding lower bound (Theorem 5).
>
> As visually summarized in Figure 3, these bounds are closely matched. For "hard" instances ($c \le 1$), both are linear. For "easy" instances ($c > 1$), both are sublinear and exhibit a similar dependence on $c$. This strong correspondence demonstrates that our analysis accurately captures the intrinsic problem difficulty and confirms the near-optimality of CRUCB.
>
> To our knowledge, this is the first work in the rising bandit literature to provide such an explicit, side-by-side comparison of upper and lower bounds as a function of the growth rate, which we believe is a significant contribution.

---

> > ### Author Response · Authors · 2025-11-17
> >
> > > **Weakness 3 & Question 3** The role of window sizes and their effect on performance is not discussed. Providing guidance or theoretical justification for the choice of window size would strengthen the practical relevance of the method.
> > The discussion on effective window size remains insufficient. In particular, the choices of window parameters for equations (10), (11), and (12) are not explained, and it is unclear how these influence theoretical guarantees or empirical performance.
> >
> > We agree that a more detailed discussion on the role of the window size parameter is crucial. We have now added a dedicated paragraph to clarify its role and the associated tradeoffs in Section 5 and 6, which we summarize here.
> >
> > Theoretically, we use an adaptive window, $h_i = \epsilon N_{i,t-1}$, growing proportionally with the number of pulls. This is crucial for balancing initial agility with long-term statistical stability. Within this design, $\epsilon$ is a hyperparameter for the bias-variance trade-off. Since it does not alter the asymptotic regret rate with respect to $T$, our analysis in Corollary 2 focused on problem difficulty rather than $\epsilon$.
> >
> > In our experiments, our method is robust to this choice. We mainly used $\epsilon=0.125$. To test sensitivity we ran additional experiments in the Path-easy environment with $\epsilon$ values of 0.05, 0.125, 0.25,  and 0.4. The results of regret are below:
> >
> > | Regret | $\epsilon=0.05$ | $\epsilon=0.125$ | $\epsilon=0.25$ | $\epsilon=0.4$ |
> > |:-------:|:--------------:|:---------------:|:---------------:|:--------------:|
> > | 100K    | 8019.34        | 8020.53         | 8020.53         | 8019.64        |
> > | 200K    | 13715.91       | 13717.10        | 13717.10        | 13716.21       |
> > | 300K    | 19060.71       | 19061.90        | 19061.90        | 19061.01       |
> >
> > ---
> >
> > > **Question 4.** Regarding the experimental setup:
> > How were the window sizes selected for the sliding-window type baselines?
> > How was R-ed-UCB (with bandit feedback) adapted to the combinatorial semi-bandit feedback model?
> >
> >
> > For the sliding-window baselines, our goal was to ensure a fair comparison against well configured versions of these algorithms. To achieve this, we tested two distinct strategies for setting the window size. The first was a fixed window size of $\sqrt{T}$, which is reported in original papers. The second was a dynamic window size of $\epsilon N_{i,t-1}$, mirroring the adaptive approach used in R-ed-UCB. For each baseline, we then reported the results from the configuration that yielded the superior performance, ensuring our comparisons are robust.
> >
> >
> > Regarding the adaptation of R-ed-UCB to the combinatorial setting, as detailed in Appendix D.1, we adapted the non-combinatorial R-ed-UCB algorithm to our setting by having it treat each super arm as a single, independent arm.
> > Specifically, instead of learning the values of base arms, the R-ed-UCB baseline applies its rising reward estimation logic directly to the observed rewards of the super arms themselves.

---

> > > ### Comment · Reviewer_zJJk · 2025-11-25
> > >
> > > I understand that in the combinatorial setting, the optimal policy may not be constant. However, it is not clearly explained how this affects the analysis compared to the previous setting. In particular:
> > >
> > > - How exactly does this difference change the analysis?
> > > - In what way is the proposed algorithm different from the earlier one?
> > > - Does the analysis still rely on Theorem 2, specifically through the quantities $B_U$ and $B_L$?
> > >
> > > If that is the case, it seems that both the algorithm and the analysis may not substantially differ from the previous setting. Could you clarify this point more explicitly?

---

> > > > ### Author Response · Authors · 2025-11-26
> > > >
> > > > Dear Reviewer zJJk,
> > > >
> > > > We sincerely thank you for your follow-up questions. We have carefully addressed each of your points below.
> > > >
> > > > ---
> > > >
> > > > > **Question 1.** How exactly does this difference change the analysis?
> > > >
> > > > We clarify that the fundamental change in the analysis stems from the non-linear reward function $r(S,\mu)$, rather than the non-constant nature of the optimal policy itself. While the previous analysis operates on linear and independent rewards, the combinatorial setting requires analyzing how estimation errors on individual base arms propagate to the super arm's reward through the non-linear function. Consequently, our analysis necessitates introducing Lipschitz continuity to bound this error propagation, a distinct analytical step completely absent in the previous framework.
> > > >
> > > > ---
> > > >
> > > > > **Question 2.** In what way is the proposed algorithm different from the earlier one?
> > > >
> > > > The difference is structural and algorithmic. Structurally, the previous rising bandit algorithm estimates parameters for each arm directly. Applying this to a combinatorial setting is intractable as it would require maintaining indices for exponentially many super arms ($2^K$). In contrast, CRUCB estimates at the base arm level ($K$) and utilizes a combinatorial Solver to construct the solution. This design is necessary to make the problem solvable but requires the distinct analysis mentioned above to handle the composition of base arm estimates.
> > > >
> > > > Algorithmically, the previous combinatorial bandit algorithm relies on an average based estimator. In contrast, CRUCB utilizes a Future-UCB index. This index explicitly incorporates the estimated growth rate (slope) to predict future potential, a component essential for rising rewards but absent in standard combinatorial methods.
> > > >
> > > > Ultimately, these two critical distinctions uniquely position CRUCB to effectively solve this problem, overcoming the fundamental limitations where existing methods fail.
> > > >
> > > >
> > > > ---
> > > >
> > > > > **Question 3.** Does the analysis still rely on Theorem 2, specifically through the quantities $B_U$ and $B_L$?
> > > >
> > > > No. We clarify that the regret upper bound analysis (Theorem 3) does not rely on the optimality result (Theorem 2). This independence is consistent with previous frameworks in both combinatorial bandits and rising bandits, where the derivation of regret bounds is typically distinct from the specific characterization of the optimal policy.
> > > >
> > > > ---
> > > >
> > > > If you have any remaining concerns or if we have misunderstood any part of your questions, please let us know. We would be happy to provide further clarifications.

---

### Official Review · Reviewer_GnsM · 2025-10-27

**Soundness:** 3
**Presentation:** 3
**Contribution:** 2
**Rating:** 4
**Confidence:** 3

**Summary:**

This work makes the first step to study the combinatorial rising bandit problem, where the mean of the chosen base arm in round $t$ will increase in round $t+1$. The authors propose the Combinatorial Rising UCB (CRUCB) algorithm, the core of which is to use a UCB constructed using the empirical mean from the latest sliding window, the exploration bonus, and the predicted improvement. Both regret upper bound and the lower bound are established for the proposed algorithm. Experiments on synthetic environments and deep reinforcement learning settings are conducted.

**Strengths:**

1. This seems to be the first work in the literature to model and resolve the combinatorial rising bandit problem.
2. The paper is generally well-written.

**Weaknesses:**

1. **Novelty**: Though the problem formulation is relatively new, the technical novelty seems somewhat limited. To me, the core part of the proposed algorithm is to use a UCB constructed using exploration bonus, the empirical mean, and the predicted improvement in the past sliding window, which is exactly the way [1] deals with rising multi-armed bandits (MABs). Of course, their method cannot deal with the combinatorial rising bandit problem. However, it is not clear to me whether there are additional technical difficulties when combining the method to deal with rising MABs and non-rising combinatorial bandits (say, [2]). As such, this prevents me from giving a positive rating for this work. And due to my relatively limited expertise in combinatorial bandits, I’d like to maintain a low level of confidence in my rating.

[1] Metelli et al. Stochastic Rising Bandits. ICML, 22.

[2] Chen et al. Combinatorial multi-armed bandit with general reward functions. NeurIPS, 16.

**Questions:**

1. As the reward of some chosen base arm $i$ in round $t$ will increase in round $t+1$, this problem seems a special case of adversarial bandits with an adaptive adversary. In this view and assuming $ r(S, \boldsymbol{\mu}) =\sum_{i\in S}\mu_i$, I think that the algorithms in previous works for adversarial combinatorial bandits with semi-bandit feedback (say, [3,4]) are applicable to this problem and that $O(\sqrt{T})$ regret upper bound is achievable. However, when $ f(n)=(n+1)^{-c}$ and $c=1.1$, this seems to contradict the lower bound of $\Omega(T^{2-c})=\Omega(T^{0.9})$. Do I miss something?

[3] Audibert et al. Regret in online combinatorial optimization. Mathematics of Operations Research, 13.

[4] Zimmert et al. Beating Stochastic and Adversarial Semi-bandits Optimally and Simultaneously. ICML, 19.

---

> ### Author Response · Authors · 2025-11-17
>
> Dear Reviewer GnsM,
>
> We are grateful for your thorough review and insightful comments. We particularly value your critical feedback regarding the technical novelty of our method, as it gives us an important opportunity to elaborate on the specific challenges that arise at the intersection of rising and combinatorial bandits. We have carefully addressed your questions and provide our responses below.
>
> ---
>
>
> > **Weakness 1. Novelty**
>
> Thank you for your detailed feedback and for giving us the opportunity to clarify the technical novelty of our work, which we view in two primary areas.
>
>
> First, our core technical novelty is the formalization and deep structural analysis of a new problem class (CRB) with a unique, emergent challenge we term **partially shared enhancement.** This is not a simple combination, as the challenge fundamentally alters the problem's structure. We prove this in Theorem 1, which shows that the optimal policy itself changes and becomes far more complex than in prior bandit settings. Critically, our further analysis in Theorem 2 then makes this problem tractable by characterizing when a simpler constant policy still serves as a good approximation. Therefore, our contribution is this deep analysis of a new and challenging problem, not a simple mix of existing methods.
>
>
> Second, a key aspect of our novelty is establishing a crucial bridge between bandit theory and its application in mainstream AI research. To do this, we are the first to frame the widely-used AntMaze benchmark as a CRB problem, demonstrating that our framework naturally models its hierarchical decision-making process. This validation in a complex, mainstream RL environment is a significant contribution because it confirms that CRB is not merely a theoretical construct but a relevant and applicable model for modern AI challenges, directly connecting our theoretical advances to the practical problems faced by the broader research community.
>
> ---
>
> > **Question 1.** As the reward of some chosen base arm $i$ in round $t$ will increase in round $t+1$, this problem seems a special case of adversarial bandits with an adaptive adversary. In this view and assuming $r(S,\boldsymbol{\mu})=\sum_{i\in S}\mu_i$, I think that the algorithms in previous works for adversarial combinatorial bandits with semi-bandit feedback (say, [3,4]) are applicable to this problem and that $O(\sqrt{T})$ regret upper bound is achievable. However, when $f(n)=(n+1)^{-c}$ and , $c=1.1$, this seems to contradict the lower bound of $\Omega(T^{2-c})=\Omega(T^{0.9})$.. Do I miss something?
>
> The apparent contradiction you've identified is not a contradiction at all, because the two regret bounds apply to fundamentally different and non-comparable notions of regret: “External regret” versus “Policy regret”.
>
> The $O(\sqrt{T})$ guarantees for adversarial bandits apply to “external regret”. In this setting, the sequence of reward functions $(r_1, ..., r_T)$ is fixed by an adversary, and the regret is defined against the best single action in hindsight:
>
> $Reg_{ext}(\pi, T) = \max_{S} \sum_{t=1}^{T} r_S(t) - \sum_{t=1}^{T} r_{S_t}(t)$
>
> In contrast, our work uses “policy regret”. Here, the reward function $\mu_t$ at time $t$ explicitly depends on the sequence of past actions $(S_1, ..., S_{t-1})$. The regret compares the algorithm's performance to the optimal dynamic policy $\pi^*$ that can strategically plan its actions to maximize long-term rewards across all possible sequences:
>
> $Reg_{pol}(\pi, T) = E_{\pi^*}\left[\sum_{t=1}^{T}r(S_t,\mu)\right] - E_{\pi}\left[\sum_{t=1}^{T}r(S_t,\mu)\right]$
>
> This is a stronger and more challenging notion of regret. External regret evaluates performance on a single path of rewards. Policy regret, however, compares the algorithm to the optimal dynamic policy $\pi^*$ , a much stronger benchmark representing an optimal plan that is effective across all possible future outcomes that its own sequence of actions can generate. A clear illustration of this fundamental gap is provided in Example 1 of (Heidari et al., ‘16), which presents a scenario where external regret is trivial while policy regret is highly non-trivial.
>
> In summary, your observation is correct that the bounds appear contradictory, but they are not directly comparable. The adversarial guarantees are for the external regret setting, whereas our lower bound of $\Omega(T^{2-c})$ is for the policy regret setting, which accurately reflects the challenge of learning in an adaptive environment.
>
> (Heidari et al. ‘16) Tight policy regret bounds for improving and decaying bandits, IJCAI, 2016

---

> > ### Author Response · Authors · 2025-11-28
> >
> > Dear Reviewer GnsM,
> >
> > We hope this message finds you well. We have incorporated your feedback into the updated paper and provided detailed answers to your questions and concerns:
> >
> > * **Novelty**: We emphasized that our contribution goes beyond combining existing methods by identifying the unique structural challenge of **partially shared enhancement**, which fundamentally alters the optimal policy. Furthermore, we demonstrated the framework's broad applicability by modeling the AntMaze benchmark as a CRB problem, effectively bridging bandit theory with modern RL applications.
> >
> > * **Comparison with adversarial bandits**: We clarified that there is no contradiction because the regret definitions differ. While adversarial bounds target external regret, our setting addresses policy regret. We have clearly distinguished between external regret and policy regret in our response.
> >
> > We would appreciate your confirmation on whether these revisions align with your expectations, or if there are further aspects you would like us to improve. We remain fully available for further discussion and look forward to your feedback.

---

### Official Review · Reviewer_hg9R · 2025-11-01

**Soundness:** 3
**Presentation:** 4
**Contribution:** 3
**Rating:** 6
**Confidence:** 4

**Summary:**

This work studies a MAB setting where it combines two existing setting Combinatorial bandits (gets selected a set of arms) in each round and the arms also gets better in each round. The setting poses an interesting challenge where in which the optimal policy has to balance out with good early pieces and late bloomers. They propose a new UCB style algorithm, CRUCB that takes into account of future reward and constructing a upper bound for each arm and according selecting the best arm. They provide regret theoretical guarantees for this algorithm complemented by experiments on different scenarios.

**Strengths:**

The work introduces a new bandits setting that combines combinatorial and rising bandits. The setting is interesting and seems to have a wide range of application.

The authors proposed a new algorithm CRUCB for this setting showcase its performance with a regret upper bound in $K$ and $L$.

The work also includes a lower bound to highlight the difficulty of this setting and to further complement the results of CRUCB's regret upper bound.

The authors provide a good experimental setup to validate CRUCB in shortest planning task and hierarchical RL tasks comparing against baselines in rising and combinatorial baselines.

**Weaknesses:**

The Solver is treated as an oracle that exist to solve this combinatorial problem and cost associated is not discussed.

Many combinatorial tasks in real world practice exhibits non-monotonic rewards but the work assumes $r$ as a monotonic function. The assumption of reward formulation being monotonic in $S$ \& $\mu$ limits its use case and generality.

*also refer questions section.

**Questions:**

1. Does CRUCB suffer any extra cost from the Solver as compared to the greedy selection of arm  ? So, by treating Solver as a oracle that exist, is there any cost that is not accounted in the regret ?

2. In case, if the $\mu_i(n+1) - \mu_i(n) = 0$, Does the setting reduce to a Standard Combinatorial algorithm ? and how does the regret compare against other existing Combinatorial algorithms ? Can you provide which term in Theorem 3 is unavoidable in comparison to those algorithm ?

3. Also, if each super arm is exclusively just the base arm and the $|S|$ = $K$, so does the setting reduces to a rising bandits then in that case as there is an assumption of concavity of $\mu_i$, how does CRUCB's regret compare against the rising bandits algorithm ? Does CRUCB recovers the regret for rising bandits ?

4. Many practical scenario involve piecewise rising (i.e. the reward rise with pulls, gets stagnant for a period and then rise), does violating Assumption 1 breaks the regret guarantee ?

5. Does Regret bound in Theorem 3 becomes sublinear if the rising effect did not become saturated ?

---

> ### Author Response · Authors · 2025-11-17
>
> Dear Reviewer hg9R,
>
> Thank you for your comprehensive review and positive assessment of our work. We are grateful for your recognition of our novel setting, theoretical contributions, and experimental validation. Your insightful questions have helped us to further refine our discussion, and we have carefully addressed each of them in the responses below.
>
> ---
>
> > **Weakness 1 & Question 1.** The Solver is treated as an oracle that exists to solve this combinatorial problem and cost associated is not discussed. Does CRUCB suffer any extra cost from the Solver as compared to the greedy selection of arm ? So, by treating Solver as an oracle that exists, is there any cost that is not accounted for in the regret ?
>
> Regarding computational cost, the standard convention in the combinatorial bandit literature is to decouple the statistical challenge of learning from the computational challenge of optimization. The role of the learning algorithm is to handle the statistical estimation of each base arm's potential value, and the Solver then uses these estimates to perform the optimization. The complexity of that oracle is thus a property of the combinatorial problem itself, not our learning algorithm.
> While this separation of concerns follows established convention, we agree that making it explicit improves clarity.  Accordingly, we have updated the manuscript to add a concise discussion of this convention and a brief summary of the solver’s computational complexity in Section 4. We hope this addresses the concern and invite the reviewer to refer to the updated PDF.
>
> ---
>
> > **Weakness 2.** Many combinatorial tasks in real world practice exhibit non-monotonic rewards but the work assumes $r$ as a monotonic function. The assumption of reward formulation being monotonic in $S$ & $\mu$  limits its use case and generality.
>
> We would like to emphasize that this is a standard and foundational assumption in the general combinatorial bandit literature (Chen et al., 2013; Wang & Chen, 2018), where it is required to make the problem tractable. This assumption holds for both linear rewards (Kveton et al., 2015) and various non-linear rewards like the $k$-MAX problem (Wang & Chen, 2023). Our work aims to introduce and solve the novel challenge of combinatorial rising rewards within this established and widely accepted framework.
>
> (Chen et al.‘13) Combinatorial Multi-Armed Bandit: General Framework and Applications, ICML, 2013
>
> (Wang & Chen‘18) Thompson Sampling for Combinatorial Semi-Bandits, ICML, 2018
>
> (Kveton et al.‘15) Tight Regret Bounds for Stochastic Combinatorial Semi-Bandits, AISTATS, 2015
>
> (Wang et al.‘23) Combinatorial Bandits for Maximum Value Reward Function under Max Value-Index Feedback, ICLR, 2024
>
> ---
>
> >**Question 2.** In case, if the $\mu_i(n+1) - \mu_i(n) = 0$, does the setting reduce to a Standard Combinatorial algorithm? And how does the regret compare against other existing Combinatorial algorithms ? Can you provide which term in Theorem 3 is unavoidable in comparison to those algorithms ?
>
> When $\mu_i(n+1)-\mu_i(n)=0$, our CRB setting reduces to a standard stationary combinatorial bandit problem. When the environment is stationary, the cumulative increment $\Upsilon(\cdot)$ in Term (i) becomes zero, However Term (ii) remains, leading to a regret of $\tilde{O}(T^{2/3})$, which reveals a gap compared to the optimal $\tilde{O}(\sqrt{T})$ rate for the stationary case. This gap arises from a fundamental and deliberate design difference in our Future-UCB index. Classical UCB-based methods use confidence bounds of the form:
>
> $\sqrt{\frac{2\log t}{N_{i,t-1}}}$,
>
> which decrease with more observations and are well-suited for stationary rewards.
> In contrast, Future-UCB index uses:
>
> $\frac{(t-N_{i,t-1}+h_i-1)}{h_i}\sqrt{\frac{10\log t^3}{h_{i}}}$,
>
> where $h_i=\epsilon N_{i,t-1}$ is the sliding window size.
> This expression is approximately $(t/N_{i,t-1})$ times larger than the classical bound, which can lead to significantly higher confidence bonuses for some arms, especially when $t$ is close to the time horizon $T$.
> CRUCB employs this significantly larger bonus to conservatively estimate future rewards. In a rising environment, relying solely on the naive empirical average would consistently underestimate an arm's long-term value.
> Our larger exploration bonus explicitly accounts for this uncertainty about future growth. While this approach is beneficial, and often necessary, in the rising setting, it leads to a degree of over-exploration in the stationary case, which is why our regret bound does not reduce to the optimal $\tilde{O}(\sqrt{T})$ rate known for classical combinatorial bandits.
>
> This is not a theoretical flaw, but rather an inherent design tradeoff reflecting CRUCB's primary objective: to robustly handle rising rewards. Developing adaptive variants that can interpolate between classical and rising-aware exploration strategies is a valuable and interesting direction for future work.

---

> > ### Author Response · Authors · 2025-11-17
> >
> > >**Question 3.** Also, if each super arm is exclusively just the base arm and the  $|S| = K$ , so does the setting reduce to a rising bandits then in that case as there is an assumption of concavity of $\mu_i$, how does CRUCB's regret compare against the rising bandits algorithm ? Does CRUCB recover the regret for rising bandits ?
> >
> > When each super arm is a singleton (i.e., $S$=\{\{$1$\},$...$,\{$K$\}\}), the combinatorial aspect vanishes, and our CRB setting indeed reduces to the standard rising bandit problem.
> > In this case, CRUCB’s behavior becomes effectively identical to existing rising bandit algorithms like R-ed-UCB (Metelli et al., 2022).
> > The Solver simplifies to a simple argmax over the $K$ arms, and our Future-UCB index functions similarly to existing estimators in that work.
> > Consequently, our regret guarantees also recover those of the standard rising bandit setting. Specifically, by setting the maximum super arm size $L=1$ in our Theorem 3, the resulting regret bound matches the known rates, demonstrating that our framework is a consistent generalization.
> >
> > (Metelli et al. ‘22) Stochastic Rising Bandit, ICML, 2022
> >
> > ---
> >
> > >**Question 4.** Many practical scenarios involve piecewise rising (i.e. the reward rises with pulls, gets stagnant for a period and then rises), does violating Assumption 1 break the regret guarantee?
> >
> > Theoretically, violating the concavity assumption (Assumption 1) does break our formal regret guarantee. This assumption is crucial as it ensures that recent progress is a reliable (i.e., optimistic) predictor of future potential. However, we would like to emphasize that this assumption, or a similar structural constraint, is fundamental to almost all rising bandit literature (Heidari et al. ‘16; Li et al. ‘20; Metelli et al. ‘22). Without some form of predictability like concavity, the problem becomes theoretically intractable. If rewards can rise arbitrarily without diminishing returns (e.g., staying flat for a long time and suddenly spiking), no algorithm could guard against this possibility without continuously exploring all arms, which would inevitably lead to linear regret. Therefore, the concavity assumption is not a weakness specific to our analysis, but rather a necessary condition to make meaningful learning design possible in a rising-reward setting.
> >
> > Empirically, we directly address the reviewer's concern by demonstrating that the piecewise rising scenario is precisely the environment of our Deep RL experiments. As shown in Figure 7a, this environment features an extended zero-reward period followed by a rising phase, and Figures 7b-c confirm CRUCB's robust outperformance. This success of CRUCB stems from its ability to effectively handle rising rewards within this combinatorial setting. Its Future-UCB index allows it to identify the potential of any base arm even after a long initial period of low rewards, as a single success gets extrapolated with the time $t$ to trigger renewed exploration. While R-ed-UCB employs a similar estimation logic, its failure highlights the critical importance of the combinatorial structure. CRUCB applies this logic to the compact set of base arms, making exploration efficient by sharing improvements. R-ed-UCB, by contrast, ignores this structure and applies the logic to each super arm independently, forcing it to inefficiently explore an exponentially larger space and fail to estimate correctly. Therefore, CRUCB's success is due to its synthesis of a rising reward mechanism with a combinatorial exploration strategy.
> >
> > (Heidari et al. ‘16) Tight policy regret bounds for improving and decaying bandits, IJCAI, 2016
> >
> > (Li et al. ‘20) Efficient Automatic CASH via Rising Bandits, AAAI, 2020
> >
> > (Metelli et al. ‘22) Stochastic Rising Bandits, ICML, 2022
> >
> > ---
> >
> >
> > >**Question 5.** Does Regret bound in Theorem 3 become sublinear if the rising effect did not become saturated ?
> >
> > Our theoretical analysis confirms that the regret bound does not remain sublinear if the rising effect does not saturate. From a learning perspective, if an arm's reward can rise persistently, an algorithm can never confidently discard it without risking linear regret.
> > This is because any currently suboptimal arm might eventually become optimal.
> >
> > Our theoretical analysis in Corollary 2 explicitly formalizes this. The corollary shows that the regret bound is directly tied to the decay rate of the reward increment function, $\Upsilon(\cdot)$ (upper-bounded by $f(n)$). As detailed in our analysis (Equation 11 in the paper), if the reward growth $f(n)$ decays too slowly such as $(n+1)^{-c}$ with $c ≤ 1$, which represents a not saturating or slowly saturating regime, our regret bound becomes linear, $O(T)$. Therefore, this linear regret is not a limitation of our algorithm but rather reflects the inherent statistical difficulty of the problem.

---

> > > ### Author Response · Authors · 2025-11-28
> > >
> > > Dear Reviewer hg9R,
> > >
> > > We hope this message finds you well. We have incorporated your feedback into the updated paper and provided detailed answers to your questions and concerns:
> > >
> > >
> > > * **Oracle cost**: We clarified that assuming oracle follows the standard convention in combinatorial bandit literature, ensuring our framework is consistent with existing researches.
> > >
> > > * **Assumptions**: We clarified that monotonicity and concavity are essential theoretical assumptions in the standard combinatorial and rising literature, respectively. They are required to ensure the problem remains tractable.
> > >
> > > * **Relationship with existing settings**: We clearly explained how our CRB setting and regret bounds relate to standard combinatorial and rising settings.
> > >
> > > * **Regret in non-saturating scenarios**: We clarified the change of the regret bound when the rising curve does not saturate, discussing the inherent trade-offs involved.
> > >
> > > We would appreciate your confirmation on whether these revisions align with your expectations, or if there are further aspects you would like us to improve. We remain fully available for further discussion and look forward to your feedback.

---

### Official Review · Reviewer_obpT · 2025-11-01

**Soundness:** 4
**Presentation:** 3
**Contribution:** 3
**Rating:** 6
**Confidence:** 3

**Summary:**

The authors propose a combinatorial rising bandit algorithm that handles cases where the expected reward of each base arm increases over time in a combinatorial setting. The paper is theoretically solid, providing both regret lower and upper bounds based on the proposed UCB-based algorithm. Simulations and the shortest-path problem both demonstrate the performance of the proposed approach. Overall, I think this is a well-written paper with enough detail to support its claims.

**Strengths:**

1. The paper is well structured, with a clear flow and proper introduction of concepts, making it easy to follow even for readers who may not work directly in this area.
2. The theoretical results are sound. I think this problem is genuinely challenging given both the combinatorial nature and the non-stationary reward that changes as arms are pulled over time. The authors set a good pace to introduce the theoretical results.
3. The simulation part is clear and effectively supports the theoretical findings.

**Weaknesses:**

See Questions below.

**Questions:**

1. I’m curious whether the authors have considered extending the framework to a Thompson Sampling (TS)-based algorithm under this setting. I saw that the simulations include TS baselines. While I understand that CRUCB should perform better since it is specifically designed to handle both the combinatorial and rising-reward aspects, I’d be interested to hear how the authors view the challenges of extending this to a TS version, since TS often slightly outperforms UCB in many settings.
2. In Theorem 3, I don’t quite understand the meaning of q. Does the theorem hold for any q \in [0,1], or is it a pre-specified value determined from upstream? How is q decided in practice?
3. What is the general computational complexity of the algorithm? I’m curious which part dominates the computation time: is it the combinatorial optimization part (the solver in CRUCB) or the rising reward evaluation part (Future-UCB calculation)? Either a theoretical complexity order or an empirical runtime illustration would be helpful.
4. It would also be nice if the authors could provide more examples or motivating scenarios showing where combinatorial rising bandit problems are commonly encountered.

---

> ### Author Response · Authors · 2025-11-17
>
> Dear Reviewer obpT,
>
> We are very grateful for your thorough and insightful review of our work. Your thoughtful comments have been invaluable in helping us refine the paper, and we appreciate the considerable time and effort you dedicated to our submission. We have carefully considered each of your points and provide our detailed responses below.
>
> ---
>
> > **Question 1.** I’m curious whether the authors have considered extending the framework to a Thompson Sampling (TS)-based algorithm under this setting. I saw that the simulations include TS baselines. While I understand that CRUCB should perform better since it is specifically designed to handle both the combinatorial and rising-reward aspects, I’d be interested to hear how the authors view the challenges of extending this to a TS version, since TS often slightly outperforms UCB in many settings.
>
> That is a fascinating direction for future work. We note that adapting Thompson Sampling (TS) to either rising bandits or combinatorial bandits alone has already proven to be a significant challenge in recent literature (Fiandri et al., 2025; Zhang et al., ‘24). The primary challenge for TS in our CRB setting is how to correctly update the posteriors of multiple base arms from a single super arm reward, especially when their mean rewards are constantly rising.
>
>
> A promising approach could be to incorporate the logic of our Future-UCB index into the sampling process itself. For instance, one might sample a mean reward from a conventional posterior, and then add a projected growth term, effectively creating a future-aware sample. This would translate the belief about the present into an actionable value that accounts for future potential.
>
> However, the theoretical analysis of such an algorithm would be a substantial undertaking. This projected sampling process is not a true Bayesian update, which would introduce a bias that is difficult to analyze, especially under the complex dependencies of the combinatorial setting. While this is a formidable challenge, we believe it is a very promising direction for future work.
>
> (Fiandri et al. ‘25) Thompson Sampling-like Algorithms for Stochastic Rising Bandits, arXiv, 2025
>
> (Zhang et al. ‘24) Thompson Sampling For Combinatorial Bandits: Polynomial Regret and Mismatched Sampling Paradox, NeurIPS, 2024
>
>
> ---
>
> > **Question 2.** In Theorem 3, I don’t quite understand the meaning of $q$. Does the theorem hold for any $q \in [0,1]$, or is it a pre-specified value determined from upstream? How is $q$ decided in practice?
>
> We clarify that $q$ is not a hyperparameter of the CRUCB algorithm, but a purely analytical tool used in our proof. Its purpose is to provide a single, unified regret bound across diverse scenarios.
> The algorithm's implementation is completely independent of $q$, and our theorem guarantees its performance uniformly. We have added a clarification to Section 5.1 to make this point explicit.

---

> > ### Author Response · Authors · 2025-11-17
> >
> > > **Question 3.** What is the general computational complexity of the algorithm? I’m curious which part dominates the computation time: is it the combinatorial optimization part (the solver in CRUCB) or the rising reward evaluation part (Future-UCB calculation)? Either a theoretical complexity order or an empirical runtime illustration would be helpful.
> >
> > The per-step complexity of CRUCB can be analyzed in two parts: the Future-UCB index calculation and the combinatorial optimization Solver. The calculation of the Future-UCB indices is highly efficient, requiring only $O(K)$ cost for the $K$ base arms. The computationally dominant part is the Solver, whose complexity is not determined by our learning algorithm but by the intrinsic structure of the underlying combinatorial problem. For example, in the online shortest path problem, the Solver can be instantiated as Dijkstra’s algorithm, with a complexity of $O(K\log K)$.
> > We would like to emphasize that, in the context of combinatorial bandits, the primary focus typically lies on the sample efficiency of the learning algorithm, while the computational cost of the Solver is treated as a separate concern. Consistent with this convention, our analysis centers on regret minimization, assuming access to an appropriate Solver for each task. We believe that clarifying this distinction improves the paper's completeness, and we have now added a brief discussion on computational complexity to the manuscript to make this explicit.
> >
> > ---
> >
> > > **Question 4.** It would also be nice if the authors could provide more examples or motivating scenarios showing where combinatorial rising bandit problems are commonly encountered.
> >
> > We agree that providing a rich set of motivating scenarios is crucial for illustrating the broad applicability of the CRB framework.
> > We would like to gently note that we have included some real-world applications, such as network routing and crowdsourcing, in Appendix A.
> > However, we recognize that expanding on these and adding more diverse examples would significantly strengthen the paper.
> > As an additional example not covered there, we highlight LLM planning: modern agentic systems repeatedly reuse a small set of tools/skills (e.g., retrieval, code execution, verification, sub-goal decomposition).
> > Here, base arms are the reusable tools/skills and a super arm is the composed plan (a chain or DAG of tools). Repeated invocation yields rising effectiveness via warm-starts and adaptation (prompt caching, retrieval-key refinement, few-shot priming), while the same tools are shared across many plans, so improvements partially transfer, which precisely matches the CRB structure.
> > These domains align with our assumptions (monotone, often concave improvement with saturation) and illustrate why CRB is a natural model for structured learning with reuse and progression. To reflect this, we have now updated Appendix A with a discussion of this LLM planning scenario.

---

> ### Author Response · Authors · 2025-11-28
>
> Dear Reviewer obpT,
>
> We hope this message finds you well. We have incorporated your feedback into the updated paper and provided detailed answers to your questions and concerns:
>
>
> * **Extension to Thompson Sampling**: We acknowledged this as a promising future direction and discussed the theoretical challenges regarding posterior updates in the CRB setting.
>
> * **Parameter $q$ in Theorem 3**: We clarified that $q$ serves purely as an analytical tool for the proof. It has no influence on the algorithm’s operation or the experimental results.
>
> * **Computational complexity**: We added a discussion clarifying that the runtime is dominated by the combinatorial Solver (oracle), while our Future-UCB index calculation remains efficient at $O(K)$
>
> * **Motivating scenarios**: We expanded Appendix A to include a new real-world application in LLM Planning (agentic systems with tool reuse), in addition to the existing examples like network routing.
>
> We would appreciate your confirmation on whether these revisions align with your expectations, or if there are further aspects you would like us to improve. We remain fully available for further discussion and look forward to your feedback.

---

### Author Response · Authors · 2025-12-04

We sincerely thank the reviewers for their constructive feedback and the Area Chair for their  time and efforts. We respectfully note that reviewers obpT and hg9R, who expressed higher confidence in their assessments (Confidence 3 and 4, respectively), rated our work favorably (Score 6), recognizing the theoretical soundness and the wide applicability of our research.

Regarding the concerns raised by reviewers GnsM and zJJk (Score 4, Confidence 3 and 2, respectively), we have provided detailed clarifications, particularly on technical novelty to directly address the questions raised in their initial reviews.

Below, we summarize the strengths recognized by the reviewers and how we clarified their initial concerns.

---

**Strengths**

*   **First work to model Combinatorial Rising Bandits (All Reviewers)**.
Reviewers unanimously acknowledge that this is the first work to formally model and resolve the Combinatorial Rising Bandit (CRB) problem, a setting where rewards increase over time in a combinatorial framework.

*   **Theoretically solid with sound results (obpT, hg9R, GnsM)**.
Reviewers highlight that the paper is theoretically well-grounded. Reviewer obpT notes the results are "sound" and the problem is "genuinely challenging," while reviewer hg9R appreciated the comprehensive inclusion of both regret upper and lower bounds.

*   **Bridging theory of CRB and hierarchical reinforcement learning applications (obpT, hg9R).**
Reviewers praised the AntMaze experiments as a significant strength. This validation demonstrates the framework's applicability beyond synthetic settings, effectively bridging theoretical bandit guarantees with practical reinforcement learning challenges.

---

**Key Clarifications and Improvements**

*   **Technical novelty (GnsM, zJJk).**
Reviewers questioned the technical novelty, specifically asking whether our approach is merely a simple combination of two existing settings (rising bandits and combinatorial bandits). In response, we clarified that our work extends beyond a simple combination by identifying the unique structural challenge of **partially shared enhancement,** which necessitates a fundamentally different analysis (Theorems 1 & 2) compared to previous settings. (line 50 ~ line 89 and Figure 1)

*   **Comparison with adversarial bandits (GnsM).**
We addressed the concern regarding the apparent contradiction between the adversarial upper bound ($O(\sqrt{T})$) and our lower bound ($\Omega(T^{2-c})$). We clarified this stems from a technical misunderstanding of regret definitions: our setting requires policy regret (vs. external regret) since the reward evolution is driven by the agent's past choices, rendering the bounds incomparable.

*   **Solver and computational complexity (obpT, hg9R).**
Following standard conventions, we clarified that learning efficiency is decoupled from the oracle's cost, which is problem-specific (e.g., Dijkstra for shortest path). To demonstrate versatility across different solvers, we highlighted experiments on MST and Matching in Appendix F.

*   **Validity of assumptions and robustness (hg9R).**
We clarified that concavity is a standard assumption in rising bandit literature for tractability. Moreover, our AntMaze experiments demonstrate strong empirical robustness, showing our algorithm outperforms baselines even when this assumption is violated.

*   **Tightness of regret bounds (zJJk).**
We explicitly showed that our upper and lower bounds match across different problem difficulties (parameterized by growth rate $c$), as summarized in our added discussion and Figure 3, confirming the near-optimality of CRUCB.

*   **Expanded discussions and additional experiments (obpT, zJJk).** we added the following experiments and discussions:
    *   **New motivating scenario (LLM Planning):** We expanded Appendix A to include LLM Planning (agentic systems with tool reuse) as a real-world application.
    *   **Hyperparameter sensitivity:** We provided additional sensitivity analysis showing that our algorithm is robust to the window size parameter $\epsilon$, maintaining stable performance across a wide range of values.

---

We hope this summary is helpful, and we sincerely thank you again for your time and consideration.

---

### Meta-Review · Area_Chair_HytM · 2026-01-09

**Summary:**

The paper introduces "Combinatorial Rising Bandits" (CRB), a new framework where an agent selects a combination of actions (super arms), and the rewards for the underlying base arms increase with usage (e.g., a robot improving through practice). The authors propose an algorithm called CRUCB and provide theoretical regret bounds and empirical validation using Deep RL environments (AntMaze).

Two reviewers (obpT, hg9R) praised the work for sound theory, novelty in defining the CRB setting, and strong experimental results.
Two reviewers (GnsM, zJJk) questioned the technical novelty, asking if the method was simply a combination of existing "Combinatorial" and "Rising" bandit techniques. They also raised concerns about regret bound definitions and hyperparameter sensitivity.

The authors argued that the problem introduces a unique challenge ("partially shared enhancement") that prevents simple combinations of existing methods. They clarified confusion regarding adversarial vs. policy regret, added a new "LLM Planning" application scenario, and demonstrated that their regret bounds are tight.

Given the soundness in theory and experiments and novelty in the setting, the AC recommends Accept.

**Reviewer Concerns:**

- zJJk questioned if the paper was just a trivial combination of "Rising Bandits" and "Combinatorial Bandits." and remained concerned after the first rebuttal. The authors provide further details about the differences between the new setting and previous settings. The AC accepts that.

- The authors provide a detailed rebuttal to other concerns of the reviewers. The AC believes these rebuttals are sufficient.

**Reviewer Scores:**

Reviewer GnsM (4 -> 4/6): I believe this reviewer would have raised their score. Their main concern was a theoretical contradiction (Adversarial vs. Policy regret), which the authors definitively proved was a misunderstanding on the reviewer's part.
Reviewer zJJk (4 -> 4/6): This reviewer remained skeptical about novelty but did not respond to the final technical clarification regarding non-linear reward propagation. I believe if they had engaged with that final detail, they would have seen the distinctness of the analysis and raised their score.
Reviewers obpT and hg9R (6 -> 6): These reviewers were already positive and their minor questions were addressed; they likely would have maintained or slightly increased their scores.

---

### Decision · Program_Chairs · 2026-01-26

Accept (Poster)